# Infinite-Width Limit of a Single Attention Layer: Analysis via Tensor Programs

**Mana Sakai**[1,3]   **Ryo Karakida**[2,3]   **Masaaki Imaizumi**[1,3]

[1]The University of Tokyo
[2]National Institute of Advanced Industrial Science and Technology
[3]RIKEN Center for Advanced Intelligence Project

mana.sakai.77@gmail.com, karakida.ryo@aist.go.jp,
imaizumi@g.ecc.u-tokyo.ac.jp

## Abstract

In modern theoretical analyses of neural networks, the infinite-width limit is often invoked to justify Gaussian approximations of neuron preactivations (e.g., via neural network Gaussian processes or Tensor Programs). However, these Gaussian-based asymptotic theories have so far been unable to capture the behavior of attention layers, except under special regimes such as infinitely many heads or tailored scaling schemes. In this paper, leveraging the Tensor Programs framework, we rigorously identify the infinite-width limit distribution of variables within a single attention layer under realistic architectural dimensionality and standard $1/\sqrt{n}$-scaling with $n$ dimensionality. We derive the exact form of this limit law without resorting to infinite-head approximations or tailored scalings, demonstrating that it departs fundamentally from Gaussianity. This limiting distribution exhibits non-Gaussianity from a hierarchical structure, being Gaussian conditional on the random similarity scores. Numerical experiments validate our theoretical predictions, confirming the effectiveness of our theory at finite width and accurate description of finite-head attentions. Beyond characterizing a standalone attention layer, our findings lay the groundwork for developing a unified theory of deep Transformer architectures in the infinite-width regime.

## 1 Introduction

A useful approach to understanding the complex probabilistic behavior of neural networks is through the study of parameter distributions in the infinite-width limit. Notable examples include the neural network Gaussian process (NNGP) [LBN+18, HBSDN20], which approximates the limit of stochastic parameter distributions with Gaussian processes; the neural tangent kernel (NTK) [JGH18], which represents the model near the initial value with kernel functions; mean field theory, which describes the update of parameter distributions [MMN18]; and the Tensor Program, developed by [Yan19a, Yan20a, YL21, Yan20b, YH21, YL23, YHB+21, YYZH24], which is a general probabilistic analytical framework that unifies the representation of the infinite-width limit for a wide range of neural architectures and multiple layers. These methods provide probabilistic models that closely approximate the complex phenomena of neural networks.

One challenge in the studies is to properly represent the attention mechanisms used in Transformers [VSP+17], which frequently appear in recent large-scale architectures [DCLT19, AAA+23, ZZL+23]. Unlike ordinary multi-layer perceptrons, an attention layer has interactions between query and key variables by multiplication, which makes the infinite-width limit distribution considerably

more complex as shown in [HBSDN20], for example. To avoid this difficulty, the existing studies have mainly limited themselves to two special settings to compute the parameter distribution of the limit: (i) *Infinite-head regime*: [HBSDN20] considers the NNGP for the attention mechanism with infinite heads, resulting in a Gaussian approximation. (ii) $1/n$-*scaling regime*: the Tensor Programs [Yan19b] approximate the multiplication by changing the scale of the input variables for attention layers from $1/\sqrt{n}$ to $1/n$, where $n$ is a dimension of variables. However, these simplifications compromise the expressiveness and structure of actual attention mechanisms (and consequently, of Transformers). Specifically, the limit distribution at the infinite-head is not an effective approximation of the actual attention mechanisms because it differs significantly from the finite-head case. Also, the $1/n$-scaling regime makes all similarity scores converge to zero in the infinite-width limit, making the model equivalent to not measuring similarity between key and query vectors. Therefore, the limit parameter distribution of the attention mechanism is still under development.

In this study, we investigate the infinite-width limit distribution of outputs of a single attention layer under the common scaling and number of heads. To achieve this, we apply the Tensor Programs framework and analyze a new class of variables defined by the multiplication of intermediate variables, and derive a corresponding limit distribution. This new class of variables allows us to represent the dot-product score by the multiplication of keys and queries in the attention mechanism.

We summarize our contributions as follows:

- *Non-Gaussian limiting distribution*: We study a distribution of outputs of a single attention layer with the $1/\sqrt{n}$-scaling and finite heads, and demonstrate that in the infinite-width limit, the output distribution converges to a hierarchical Gaussian distribution, which is a type of non-Gaussian. Specifically, the limiting distribution is a Gaussian conditional on the random similarity score, and this score variable itself converges to a Gaussian.

- *Consistency with numerical experiments*: Our experiments justify that our theoretical limit distribution accurately captures the non-Gaussian behaviors exhibited by attention mechanisms. Specifically, even when the width is finite, our theory proves sufficiently accurate, provided that the dimension is large enough.

- *Novel proof technique with dot-products*: We develop a novel proof technique focused on analyzing the similarity score variables by the dot-products of an attention layer. More concretely, we first show a convergence of the score variables to their limiting distributions, and then prove a conditional weak convergence of the outputs of an attention layer. This analysis characterizes the output distribution of an attention layer by incorporating the intrinsic randomness from the dot-product.

## 1.1 Related Works

The concentration of measures in neural networks plays a fundamental role in both theoretical analysis and practical applications of machine learning. Initially, studies in this area aimed to characterize the feedforward signal propagation in wide neural networks with random weights, inspired by statistical mechanics [Ama77, SCS88]. The outputs of such random neural networks converge to a Gaussian process, and the computation of signal propagation reduces to the composition of kernel functions of Gaussian processes utilized in machine learning [Nea96, Wil96, DFS16, LBN+18, dGMHR+18]. They are often referred to as the NNGP. Since the NNGP kernel captures intrinsic inductive biases of architectures, its prediction performance correlates well with trained networks across different architectures [LBN+18]. Moreover, one can interpret NNGP as a network with random initialization of optimization, which naturally leads to quantitative insight into desirable weight scales to avoid exploding/vanishing signal and gradient problems [PLR+16, SGGSD17].

Depending on the network architecture, various types of kernels can be obtained, not only for fully-connected neural networks, but also convolutional neural networks (CNNs) [NXL+19], skip connections [YS17], naive or gated recurrent neural networks (RNNs) [CPS18, Yan19b], and more [YPR+19, GHLG23]. While some classical works assumed random Gaussian weights generated in an i.i.d. manner, recent research has shown that the same NNGP can be derived even in networks with non-Gaussian [GY22] or weakly correlated weights [SNT24]. The NNGP can also be obtained for a network with a narrow bottleneck layer sandwiched between wide layers [APH20].

Two pioneering studies have investigated the NNGP of self-attention layers [Yan19b, HBSDN20]. Initially, [Yan19b] pointed out that an unconventional scaling factor of $1/n$ in the softmax function

enables straightforward NNGP evaluation. Greg Yang has introduced the theoretical framework of Tensor Programs [Yan19a, Yan20a, Yan20b], systematically composing kernel functions, including NNGPs, for modern neural networks. Transformers with the $1/n$-scaling fall within the scope of the applicability of Tensor Programs. However, the $1/n$-scaling is rarely used in theoretical or practical contexts, leaving the more realistic $1/\sqrt{n}$-scaling unresolved. To attack this problem, [HBSDN20] analyzed self-attention by varying the number of heads. They numerically demonstrated non-Gaussian behavior emerging in the single-head case due to stochasticity and correlations within the attention matrix, even at infinite embedding dimensions. They further showed that if we take the infinite limit of the number of heads, Gaussian behavior emerges in the self-attention output and defines an NNGP kernel termed infinite attention. Although this infinite attention empirically improved performance on certain NNGP regression benchmarks, its suitability as a theoretical foundation for realistic self-attention remains uncertain. This is because practical attention implementations typically use only 1-128 heads [EXW⁺24], far fewer than the embedding dimension.

Beyond the classical NNGP and Tensor Programs analyses, several recent works have examined Transformers under the standard $1/\sqrt{n}$ scaling and related asymptotic or dynamical regimes. [DYZ23] analyzed Transformers by tracking the first two moments (the kernel) of the signal to characterize propagation at initialization and during training. [CNQG24] applied mean-field theory to characterize the edge of chaos via forward and backward signal propagation, assuming Gaussianity of the QK product. [BCP24] employed dynamical mean field theory to study training dynamics under various infinite limits, including infinite width, heads, and depth, identifying parameterizations that ensure stable feature learning over time. [NLL⁺23] modeled signal evolution with a stochastic differential equation, which requires modifying the softmax function.

## 2 Preliminary

### 2.1 Notation and Setup of Neural Networks

We define the notation for a standard neural network and its usage. In what follows, we denote by $n$ the dimensionality corresponding to the network's width. Although we can vary $n$ across different layers or architectures, for simplicity we here treat every layer as having the same width $n$. A comprehensive summary of the notations used throughout this paper is provided in Appendix A.

**Neural network** We define notation for standard neural networks, excluding the attention mechanism. In particular, we adopt notations inspired by the framework of Tensor Programs [Yan19a, Yan19b], which allow us to describe a broad class of neural network architectures.

We describe an architecture of feed-forward neural networks as a finite set of $\mathbb{R}^n$-valued random vectors $h^1, \ldots, h^J$, which is inductively generated by the following rule. We fix a nonempty subset $\mathcal{V}_0 \subset \{h^1, \ldots, h^J\}$, called the set of initial vectors (input layer). For each index $k$ with $h^k \notin \mathcal{V}_0$, the vector $h^k$ is generated either by matrix multiplication (MatMul) or by a coordinatewise nonlinearity (Nonlin). In the MatMul rule, given a weight matrix $W \in \mathbb{R}^{n \times n}$ and some $j \neq k$, one sets $h^k = Wh^j$. In the Nonlin rule, given $k \notin \{j_1, \ldots, j_m\} \subset [J]$ and a function $\phi : \mathbb{R}^m \to \mathbb{R}$, one sets

$$h^k = \phi(h^{j_1}, \ldots, h^{j_m}), \quad h^k_\alpha = \phi(h^{j_1}_\alpha, \ldots, h^{j_m}_\alpha) \quad (\alpha \in [n]).$$

This type of Tensor Program is called NETSOR, and it covers a broad range of neural network architectures, including a perceptron layer, a convolutional layer, a recurrent layer, and many others [Yan19a, Yan19b].

We present the perceptron layer as a specific neural network represented by NETSOR. Suppose $x^{\ell-1} \in \mathbb{R}^n$ is the input of the $\ell$-th layer. It generates pre-activation variable $z^\ell \in \mathbb{R}^n$ and the input of the next layer $x^\ell \in \mathbb{R}^n$ by

$$z^\ell = W^\ell x^{\ell-1}, \quad x^\ell = \phi(z^\ell),$$

where $W^\ell \in \mathbb{R}^{n \times n}$ is a weight matrix and $\phi : \mathbb{R} \to \mathbb{R}$ is a (potentially nonlinear) coordinatewise activation function. The vectors $z^\ell$ and $x^\ell$ are generated by MatMul and Nonlin, respectively.

**Multi-head attention** We define the attention layer. Let $s$ and $H$ denote the spatial dimension and the number of heads, respectively. Suppose $W^{Q,a}, W^{K,a}, W^{V,a}, W^{O,a} \in \mathbb{R}^{n \times n}$ are weight

matrices for head $a \in [H]$. With an input sequence of $s$ random vectors $x^1, \ldots, x^s \in \mathbb{R}^n$, we define $X \in \mathbb{R}^{s \times n}$ as a matrix whose $i$-th row is $x^i$, i.e., $X^\top = [x^1 \cdots x^s]$. For each head $a \in [H]$, define $Q^{(a)} = X(W^{Q,a})^\top, K^{(a)} = X(W^{K,a})^\top, V^{(a)} = X(W^{V,a})^\top$, so that $Q^{(a)}, K^{(a)}, V^{(a)}$ are $\mathbb{R}^{s \times n}$ matrices. The scaled dot-product score matrix is given by

$$G^{(a)} = \frac{1}{\sqrt{n}} Q^{(a)} (K^{(a)})^\top = (p_{i,j}^{(a)})_{i,j \in [s]} \in \mathbb{R}^{s \times s}, \quad p_{i,j}^{(a)} = \frac{1}{\sqrt{n}} (W^{Q,a} x^i)^\top (W^{K,a} x^j). \quad (1)$$

Here, the common scaling $1/\sqrt{n}$ in the definition of $p_{i,j}^{(a)}$ is a key issue in this study.[1] Applying the row-wise softmax function to the score matrix, we define

$$A^{(a)} = \mathrm{SoftMax}(G^{(a)}) \in \mathbb{R}^{s \times s}, \quad A_{i,j}^{(a)} = \mathrm{SoftMax}_j(p_{i,1}^{(a)}, \ldots, p_{i,s}^{(a)}).$$

We then weight the values to form each head's output $\mathrm{Head}^{(a)} = A^{(a)} V^{(a)} \in \mathbb{R}^{s \times n}$. Finally, we sum across all $H$ heads to recover the full attention output $\mathrm{MultiHead} = H^{-\frac{1}{2}} \sum_{a=1}^{H} \mathrm{Head}^{(a)} (W^{O,a})^\top \in \mathbb{R}^{s \times n}$, where each row $\mathrm{MultiHead}_{i\cdot}$ is given by

$$(\mathrm{MultiHead}_{i\cdot})^\top = \frac{1}{\sqrt{H}} \sum_{a=1}^{H} \sum_{j=1}^{s} W^{O,a} W^{V,a} x^i \mathrm{SoftMax}_j(p_{i,1}^{(a)}, \ldots, p_{i,s}^{(a)}) \in \mathbb{R}^n \quad (i \in [s]). \quad (2)$$

**Weight initialization**   We set the weight matrices such as $W^\ell, W^{Q,a}, W^{K,a}, W^{V,a}$, and $W^{O,a}$, and initial vectors $h \in \mathcal{V}_0$ as follows:

(i) Each weight matrix $W$ is independent. Each $(\alpha, \beta)$ element of $W$ is sampled i.i.d. from $W_{\alpha\beta} \sim N(0, \sigma_W^2/n)$, where $\sigma_W > 0$ is a constant that may depend on $W$.

(ii) Let $Z^{\mathcal{V}_0} = \{Z^h : h \in \mathcal{V}_0\} \in \mathbb{R}^{|\mathcal{V}_0|}$ be a multivariate normal distribution. For each $\alpha \in [n]$, the collection of $\alpha$-th components of all initial vectors in $\mathcal{V}_0$, denoted by $\{h_\alpha : h \in \mathcal{V}_0\}$, is sampled i.i.d. from $Z^{\mathcal{V}_0}$.

This setup of weight matrices is common in practice (see [GB10] for summary) and also identical to that in the existing Tensor Programs [Yan19a, Yan19b, Yan20b].

## 2.2   Distribution of Attention Mechanism in Previous Setup

We discuss the existing challenges on characterizing the output distribution of the attention layer, which exhibits non-Gaussian behavior. To derive this distribution, two regimes are typically considered: the choice of scaling in the similarity computation, and the number of heads.

$1/n$**-scaling regime**   [Yan19b] studies an attention layer whose scaling term $1/\sqrt{n}$ in the scaled dot-product of Eq. (1) is replaced by $1/n$. In this regime, the Tensor Programs framework can show that the outputs of the attention layer converge to a Gaussian distribution: as $n \to \infty$, it holds that

$$\mathrm{MultiHead}_{i\alpha} \xrightarrow{d} N(0, \kappa^2), \quad (\alpha \in [n]),$$

with some $\kappa > 0$ (see Appendix A and Theorem E.8 in [Yan19b] for details). Intuitively, in the $1/n$-scaling regime, the dot-product score $p_{i,j}^{(a)}$ converges to 0 for *every* pair $(i, j) \in [s]^2$ and head $a \in [H]$, which simplifies the non-Gaussian behavior of the attention outputs.

**Infinite-heads regime**   [HBSDN20] considers the regime in which many dimensionalies diverge to infinite, then shows the convergence of the attention outputs to Gaussian. Specifically, it is shown that as $n, H \to \infty$, it holds that

$$\mathrm{MultiHead}_{i\alpha} \xrightarrow{d} N(0, (\kappa')^2), \quad (\alpha \in [n]),$$

---

[1]Our analysis focuses on the $1/\sqrt{n}$-scaling, a choice motivated by its prevalence in both practical Transformer implementations and the theoretical literature (e.g., [HBSDN20, DYZ23, CNQG24]). While some analyses have raised questions about its stability [YHB+21, BCP24], the precise conditions for stable training remain an active area of research and may depend on factors often abstracted away in simplified limits, such as the number of tokens or data-specific statistics. For instance, recent work has shown that data statistics can significantly alter optimal scaling rules [HL25].

with some $\kappa' > 0$ (see Theorem 3 in [HBSDN20]). The variance $(\kappa')^2$ is described by a covariance of the nonlinearly transformed dot-product score $p_{i,j}^{(a)}$. In the result, by letting the number of heads $H$ grow to infinite, the complex non-Gaussian effects from the dot-product score are smoothed out, yielding a convenient Gaussian distribution.

While these approximations are analytically appealing, they have notable limitations. In practice, Transformers typically use $1/\sqrt{n}$-scaling (see Equation (1) in [VSP+17]) and a finite number of heads, resulting in the frequently observed non-Gaussian behaviors of attention layers. Although the above regimes simplify the behavior into a Gaussian form for tractability, a more precise theory is needed to capture the true behavior of attention layers in practice.

## 3 Main Theorem

### 3.1 Limiting Distribution

We introduce the limiting distribution for two types of quantities: variables generated within the NETSOR program, and the scalar dot-products that arise in attention mechanisms. The convergence of these variables to their limiting distributions is formally established in our main result, Theorem 3.1.

**Definition 3.1** (Limiting Distribution). **(A) Limiting distribution for vectors in the NETSOR program:** For each vector $h \in \mathbb{R}^n$ in the NETSOR program, there exist a corresponding random variable $Z^h$ as follows:

(i) If $h$ is an initial vector from $\mathcal{V}_0$, then $Z^h$ follows the distribution specified in Section 2.1.

(ii) If $g^1, \ldots, g^k$ are generated by MatMul, then $(Z^{g^1}, \ldots, Z^{g^k})$ is a zero-mean Gaussian vector. Specifically, for $g^i = W^i h^i$ and $g^j = W^j h^j$, their covariance is given by

$$\mathrm{Cov}(Z^{g^i}, Z^{g^j}) = \begin{cases} 0 & (\text{ if } W^i \text{ and } W^j \text{ are different matrices}), \\ \sigma_W^2 \mathbb{E}[Z^{h^i} Z^{h^j}] & (\text{ if } W^i \text{ and } W^j \text{ are the same matrix}). \end{cases}$$

(iii) If $h$ is generated by Nonlin, $h = \phi(h^1, \ldots, h^k)$, then $Z^h$ is defined as $Z^h = \phi(Z^{h^1}, \ldots, Z^{h^k})$.

**(B) Limiting distribution for scalar dot-products:** For each scalar $p_i = (g^{i,1})^\top g^{i,2}/\sqrt{n}$, where $g^{i,1}$ and $g^{i,2}$ are outputs of MatMul within the NETSOR program, we define $\mathring{p}_i$ as follows:

(iv) $(\mathring{p}_1, \ldots, \mathring{p}_r)$ is a zero-mean Gaussian vector. This vector is statistically independent of all $Z^h$ variables defined in (A). The covariance of $(\mathring{p}_1, \ldots, \mathring{p}_r)$ is given by

$$\mathrm{Cov}(\mathring{p}_i, \mathring{p}_k) = \mathbb{E}[Z^{g^{i,1}} Z^{g^{i,2}} Z^{g^{k,1}} Z^{g^{k,2}}].$$

In Definition 3.1, the formulations for the limiting random variables $Z^h$ associated with initial vectors, MatMul outputs, and Nonlin outputs are developed by the existing Tensor Programs framework [Yan19b]. Specifically, the limiting distribution of initial vectors follows that given in Section 2.1; variables generated by MatMul converge to Gaussian distribution; and variables generated by Nonlin are nonlinear transformations of the corresponding limiting distributions.

The distinct aspect of our analysis lies in the treatment of the limiting distribution for the scalar dot-products $(\mathring{p}_1, \ldots, \mathring{p}_r)$, which represent pre-softmax attention scores. While the marginal Gaussian distribution of these scores is consistent with that in the infinite-head limit (see Theorem 3 in [HBSDN20]), our framework explicitly defines them as a Gaussian vector that is *independent* of the limiting distributions $Z^h$ within the NETSOR program. This independence facilitates the computation of the limiting distribution of the attention outputs, as it allows us to treat the attention scores separately from the other variables in the NETSOR program.

### 3.2 Statement

In this section, we show the convergence of the distribution of variables in the presence of an attention layer as the main result. Here, we refer to a function as pseudo-Lipschitz if it is pseudo-Lipschitz of order $d$ for some $d \in [2, \infty)$. See Definition C.1 for the definition of pseudo-Lipschitz functions.

As a preparation, we first introduce the results of the convergence of the NETSOR program without an attention layer. The following theorem is a slight simplification of Theorem 5.4 in [Yan19b].[2]

**Fact 3.1** (NETSOR Master Theorem [Yan19b]). *Consider a NETSOR program. Suppose all initial vectors and weight matrices are sampled as explained in Section 2.1, and all nonlinearities used in* Nonlin *are pseudo-Lipschitz. For any positive integer $k$, let $h^1, \ldots, h^k$ be any vectors in the NETSOR program. Then, for any pseudo-Lipschitz function $\psi : \mathbb{R}^k \to \mathbb{R}$, we have*

$$\frac{1}{n} \sum_{\alpha=1}^{n} \psi(h^1, \ldots, h^k) \xrightarrow{a.s.} \mathbb{E}[\psi(Z^{h^1}, \ldots, Z^{h^k})]$$

*as $n \to \infty$. Here, $Z^{h^1}, \ldots, Z^{h^k}$ are defined in Definition 3.1, (i)–(iii).*

Building on the NETSOR master theorem, we now present our main result. Before that, we introduce some assumptions specific to the attention layer.

**Assumption 3.1.** *Let $r$ and $m$ be positive integers that satisfy $m \geq 2r$. Suppose $g^1, \ldots, g^m \in \mathbb{R}^n$ are vectors in the NETSOR program generated by* MatMul. *We assume that a subset $\{g^{i,j} : i \in [r], \ j \in [2]\} \subset \{g^1, \ldots, g^m\}$ can be expressed as, without loss of generality,[3]*

$$g^{i,j} = W^{i,j} x^{i,j}, \quad x^{i,j} = \phi^{i,j}(g^1, \ldots, g^m) \quad (i \in [r], \ j \in [2]),$$

*where each $\phi^{i,j}$ is a bounded and pseudo-Lipschitz function. The weight matrices $W^{i,j} \in \mathbb{R}^{n \times n}$ satisfy two conditions:*

(a) *$\{W^{i,j} : i \in [r], \ j \in [2]\}$ is specific to the generation of $\{g^{i,j} : i \in [r], \ j \in [2]\}$ and is not used for any $g \in \{g^1, \ldots, g^m\} \setminus \{g^{i,j} : i \in [r], \ j \in [2]\}$.*
(b) *It is permissible for $W^{i,j}$ to be the same matrix as $W^{i',j'}$ unless $i = i', j \neq j'$ (e.g., $W^{1,1}$ could be the same matrix as $W^{2,1}$).*

*Finally, we define the scalar dot-products $\{p_i : i \in [r]\}$ by*

$$p_i = \frac{1}{\sqrt{n}} (g^{i,1})^{\top} g^{i,2} \quad (i \in [r]).$$

**Theorem 3.1.** *Consider a NETSOR program, and suppose all nonlinearities used in* Nonlin *are pseudo-Lipschitz. We adopt the settings and notations from Assumption 3.1, which defines vectors $g^1, \ldots, g^m$ and scalar dot-products $p_1, \ldots, p_r$. Further, suppose all initial vectors and weight matrices are sampled as explained in Section 2.1. Now, let $h^1, \ldots, h^k \in \mathbb{R}^n$ be vectors whose elements are given by*

$$h_{\alpha}^{j} = \varphi^{j}(g_{\alpha}^1, \ldots, g_{\alpha}^m, p_1, \ldots, p_r) \quad (\alpha \in [n], \ j \in [k]),$$

*where each $\varphi^j$ is a pseudo-Lipschitz function. Then, for any bounded and pseudo-Lipschitz function $\psi : \mathbb{R}^k \to \mathbb{R}$, we have*

$$\frac{1}{n} \sum_{\alpha=1}^{n} \psi(h_{\alpha}^1, \ldots, h_{\alpha}^k) \xrightarrow{d} \mathbb{E}[\psi(Z^{h^1}, \ldots, Z^{h^k}) \mid \mathring{p}_1, \ldots, \mathring{p}_r]$$

*as $n \to \infty$. Here, $Z^{h^j}$ is given by $Z^{h^j} = \varphi^j(Z^{g^1}, \ldots, Z^{g^m}, \mathring{p}_1, \ldots, \mathring{p}_r)$ for $j \in [k]$, and $(Z^{g^1}, \ldots, Z^{g^m}, \mathring{p}_1, \ldots, \mathring{p}_r)$ is defined in Definition 3.1.*

It may be noted that in the statement of Theorem 3.1, the boundedness of $\phi^{i,j}$ as in Assumption 3.1 is not essential, and we can drop this condition. However, we keep it for simplicity of the proof. Also, note that in this statement, $\mathring{p}_1, \ldots, \mathring{p}_r$ are random variables with positive variance, and the convergence in distribution $\to^d$ described in the statement refers to convergence to the distribution of them. Unlike the conventional master theorem derived from Tensor Programs [Yan19a, Yan19b],

---

[2][Yan19b] considers controlled functions instead of pseudo-Lipschitz functions, the former of which is a generalization of pseudo-Lipschitz functions.

[3]The Tensor Programs framework generates vectors inductively, precluding circular dependencies (see Section 2.1). Consequently, for each $x^{i,j} = \phi^{i,j}(g^1, \ldots, g^m)$, any arguments from $\{g^1, \ldots, g^m\}$ that effectively contribute to the output $x^{i,j}$ (i.e., those upon which the function $\phi^{i,j}$ actually depends) must be defined prior to $g^{i,j} = W^{i,j} x^{i,j}$ in this inductive sequence.

this theorem simultaneously characterizes the randomness of the ordinary variables like $g^1, \ldots, g^m$, along with the effects of finite-dimensional random variables $\mathring{p}_1, \ldots, \mathring{p}_r$.

Theorem 3.1 implies that, due to the randomness of $(\mathring{p}_1, \ldots, \mathring{p}_r)$, the overall outputs resemble a hierarchical distribution, where the variance of one limiting distribution may depend on a realization of another limiting distribution $(\mathring{p}_1, \ldots, \mathring{p}_r)$. This result differs from existing results by the Tensor Programs where the variance depends on moments of the other distributions. This result leads to a non-Gaussian limiting distribution in the attention case, whose details will be provided in Example 3.1.

The following corollary is an analogue of Theorem A.5 in [Yan20b].

**Corollary 3.2** (Coordinatewise Convergence)**.** *Assume the same premise as in Theorem 3.1. Then, for all $\alpha \geq 1$, we have*

$$(h_\alpha^1, \ldots, h_\alpha^k) \xrightarrow{d} (Z^{h^1}, \ldots, Z^{h^k}).$$

We present specific multi-head attention results as an example of the application of Corollary 3.2. This example implies that the output of the attention mechanisms is described by a non-Gaussian distribution. Specifically, when $y^i$ is an output of a multi-head attention layer as MultiHead$_i$. defined in Eq. (2), $(Z^{y^1}, \ldots, Z^{y^s})$ follows a hierarchical Gaussian distribution whose conditioning $(\mathring{p}_1, \ldots, \mathring{p}_r) = \{\mathring{p}_{i,j}^{(a)} : i, j \in [s], a \in [H]\}$ itself is a random variable, causing $(Z^{y^1}, \ldots, Z^{y^s})$ to follow a non-Gaussian distribution overall. We provide the details as follows:

**Example 3.1** (Multi-Head Attention)**.** We consider the multi-head attention in Eq. (2). Recall that $s$ is the spatial dimension and $H$ is the number of heads. We sample new weight matrices $W^{Q,a}, W^{K,a}, W^{V,a}, W^{O,a} \in \mathbb{R}^{n \times n}$ for $a \in [H]$. Let $x^1, \ldots, x^s \in \mathbb{R}^n$ be vectors within the Netsor program. We assume these input vectors are generated by Nonlin, where the nonlinearity is bounded and pseudo-Lipschitz. For $j \in [s], a \in [H]$, we define the value vectors $v^{a,j} = W^{V,a} x^j \in \mathbb{R}^n$ ($j \in [s], a \in [H]$) and its further transform $\tilde{v}^{a,j} = W^{O,a} v^{a,j} \in \mathbb{R}^n$. Finally, with the score $p_{i,j}^{(a)}$ in Eq. (1), we rewrite an element of the multi-head attention Eq. (2) as

$$y_\alpha^i = \frac{1}{\sqrt{H}} \sum_{a=1}^{H} \sum_{j=1}^{s} \text{SoftMax}_j(p_{i,1}^{(a)}, \ldots, p_{i,s}^{(a)}) \tilde{v}_\alpha^{a,j}$$

$$=: \varphi^i \left( \{\tilde{v}_\alpha^{a,j} : j \in [s], a \in [H]\}, \{p_{i,j}^{(a)} : i, j \in [s], a \in [H]\} \right),$$

by introducing functions $\varphi^i$ ($i \in [s]$), each of which is a pseudo-Lipschitz function.[4] Then, by Corollary 3.2, we have

$$(y_\alpha^1, \ldots, y_\alpha^s) \xrightarrow{d} (Z^{y^1}, \ldots, Z^{y^s}) \quad (n \to \infty),$$

where $Z^{y^i}$ is defined by $Z^{y^i} = H^{-\frac{1}{2}} \sum_{a=1}^{H} \sum_{j=1}^{s} \text{SoftMax}_j(\mathring{p}_{i,1}^{(a)}, \ldots, \mathring{p}_{i,s}^{(a)}) Z^{\tilde{v}^{a,j}}$. By Definition 3.1, $\{Z^{\tilde{v}^{a,j}} : j \in [s], a \in [H]\}$ and $\{\mathring{p}_{i,j}^{(a)} : i, j \in [s], a \in [H]\}$ satisfy the following.

- $\{Z^{\tilde{v}^{a,j}} : j \in [s], a \in [H]\}$ is independent of $\{\mathring{p}_{i,j}^{(a)} : i, j \in [s], a \in [H]\}$.
- $\{Z^{\tilde{v}^{a,j}} : j \in [s], a \in [H]\}$ is jointly Gaussian with zero mean. For $a, a' \in [H]$ and $j, j' \in [s]$, the covariance is given by

$$\text{Cov}(Z^{\tilde{v}^{a,j}}, Z^{\tilde{v}^{a',j'}}) = \begin{cases} 0 & (a \neq a'), \\ \sigma_{W^{O,a}}^2 \sigma_{W^{V,a}}^2 \mathbb{E}[Z^{x^j} Z^{x^{j'}}] & (a = a'). \end{cases}$$

- $\{\mathring{p}_{i,j}^{(a)} : i, j \in [s], a \in [H]\}$ is jointly Gaussian with zero mean. For $a, a' \in [H]$ and $i, i', j, j' \in [s]$, the covariance is given by

$$\text{Cov}(\mathring{p}_{i,j}^{(a)}, \mathring{p}_{i',j'}^{(a')}) = \begin{cases} 0 & (a \neq a'), \\ \sigma_{W^{Q,a}}^2 \sigma_{W^{K,a}}^2 \mathbb{E}[Z^{x^i} Z^{x^{i'}}] \mathbb{E}[Z^{x^j} Z^{x^{j'}}] & (a = a'). \end{cases}$$

---

[4]See Proposition C.3 and Lemma C.4. Note that SoftMax is Lipschitz so that it is also pseudo-Lipschitz (Fact C.3).

Note that $\{Z^{x^j} : j \in [s]\}$ can be computed by Definition 3.1.

The pseudocode for this example is presented in Algorithm 1 in Appendix B.1.

**Remark 3.1.** The non-Gaussianity of the limiting distribution of the attention layer highlights a significant challenge for theoretical analysis, since many existing frameworks, such as NNGP and Tensor Programs, are developed under regimes where the limiting distributions are Gaussian (or can be expressed as Gaussian components with correction terms). Extending these tools to rigorously handle such non-Gaussian behaviors remains an important open problem, and we believe our work provides a concrete starting point for such future developments.

**Remark 3.2.** Our result in Example 3.1 has a close relationship with the result of [HBSDN20]. First, the limiting distribution of the attention scores in our framework perfectly matches the Gaussian distribution in their infinite-head limit. Second, the variance of our finite-head non-Gaussian output distribution theoretically matches the variance of their infinite-head Gaussian output distribution. This alignment suggests that while the infinite-head limit correctly captures the second-order statistics of the output, our finite-head analysis provides a more complete picture by characterizing the non-Gaussian aspects of the distribution that are not captured by second-order statistics alone.

**Remark 3.3.** We can show that the limiting distribution $Z^{y^i}$ is sub-Gaussian, provided that bounded nonlinearities are used for the attention scores (e.g., the softmax function as in Example 3.1). In other words, although $Z^{y^i}$ cannot be approximated by a single Gaussian distribution with mean $\mathbb{E}(Z^{y^i})$ and variance $\mathrm{Var}(Z^{y^i})$, its tail probability decays at a Gaussian rate, possibly with a larger variance parameter. A detailed discussion is provided in Appendix E.

## 4 Proof Sketch of Theorem 3.1

We sketch the proof of Theorem 3.1. First, observe that it is equivalent to the following statement:

**Theorem 4.1.** *Assume the same premise as in Theorem 3.1. Then, for any bounded and pseudo-Lipschitz function $\psi$, we have*

$$p$$

*as $n \to \infty$, where $(Z^{g^1}, \ldots, Z^{g^m}, \mathring{p}_1, \ldots, \mathring{p}_r)$ is defined in Definition 3.1.*

Clearly, Theorem 3.1 implies Theorem 4.1. Conversely, applying Proposition C.3 establishes the opposite direction.

In what follows, we outline the proof of Theorem 4.1. Hereafter, $x_{1:k}$ denotes the vector $(x_1, \ldots, x_k)$, and likewise, $x_\alpha^{1:k}$ denotes the vector $(x_\alpha^1, \ldots, x_\alpha^k)$. Fix a bounded and pseudo-Lipschitz function $\psi$. By Lemma C.2, it suffices to show that, for any bounded and Lipschitz function $f$,

$$\left| \mathbb{E} f \left( \frac{1}{n} \sum_{\alpha=1}^n \psi(g_\alpha^{1:m}, p_{1:r}) \right) - \mathbb{E} f \left( \mathbb{E}[\psi(Z^{g^{1:m}}, \mathring{p}_{1:r}) \mid \mathring{p}_{1:r}] \right) \right| \to 0$$

holds. Note that by Fact 3.1, we have

$$\frac{1}{n} \sum_{\alpha=1}^n \tilde{\psi}(g_\alpha^{1:m}) \xrightarrow{a.s.} \mathbb{E}[\tilde{\psi}(Z^{g^{1:m}})] \tag{3}$$

for any pseudo-Lipschitz function $\tilde{\psi}$. Using this result, we first show that $p_{1:r}$ converges in distribution to $\mathring{p}_{1:r}$, which is a Gaussian vector defined by Definition 3.1 (Appendix D.1.1). Then, we consider

$$\left| \mathbb{E} f \left( \frac{1}{n} \sum_{\alpha=1}^n \psi(g_\alpha^{1:m}, p_{1:r}) \right) - \mathbb{E} f \left( \mathbb{E}[\psi(Z^{g^{1:m}}, \mathring{p}_{1:r}) \mid \mathring{p}_{1:r}] \right) \right| \le S_1 + S_2,$$

where $S_1$ and $S_2$ are given by

$$S_1 = \left| \mathbb{E} f \left( \frac{1}{n} \sum_{\alpha=1}^n \psi(g_\alpha^{1:m}, p_{1:r}) \right) - \mathbb{E} f \left( \mathbb{E} \left[ \psi(Z^{g^{1:m}}, p_{1:r}) \mid p_{1:r} \right] \right) \right|,$$

$$S_2 = \left| \mathbb{E} f \left( \mathbb{E} \left[ \psi(Z^{g^{1:m}}, p_{1:r}) \mid p_{1:r} \right] \right) - \mathbb{E} f \left( \mathbb{E}[\psi(Z^{g^{1:m}}, \mathring{p}_{1:r}) \mid \mathring{p}_{1:r}] \right) \right|.$$

We separately show $S_1 \to 0$ (Appendix D.1.2) and $S_2 \to 0$ (Appendix D.1.3).

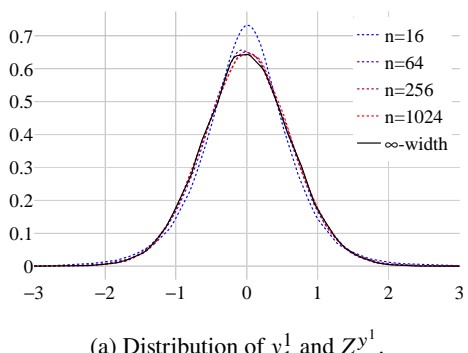
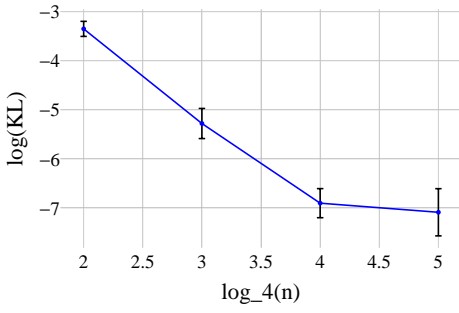

| (a) Distribution of $y_1^1$ and $Z^{y^1}$. | (b) KL divergence with error bars. |

Figure 1: Comparison of the distribution of the attention output $y_1^1$ and its infinite-width limit $Z^{y^1}$ in Example 3.1. **(a)** Kernel density estimates of the empirical distribution (via Monte Carlo sampling) of $y_1^1$ for widths $n \in \{16, 64, 256, 1024\}$ (dashed lines) alongside that of $Z^{y^1}$ (solid line), showing the convergence of the finite-width distribution to its limit. **(b)** Average of the log-KL divergence $\log \mathrm{KL}(\mathrm{Dist}(y_1^1)\|\mathrm{Dist}(Z^{y^1}))$ over 10 independent trials, plotted against $\log_4(n)$ with error bars indicating one standard deviation, confirming a decreasing trend.

# 5 Simulation and Discussion

We perform simulations to validate the infinite-width limit distributions derived in Example 3.1. Details on the simulation setting can be found in Appendix B. All simulation codes are available at https://github.com/manasakai/infinite-width-attention.

## 5.1 Effect of Finite Width

To assess how well the infinite-width theory of Theorem 3.1 aligns with finite-width multi-head attention behavior and to verify that discrepancies diminish as width grows, we perform 10 independent experiments for each width $n \in \{16, 64, 256, 1024\}$. In each trial, we draw samples of the multi-head output $y_1^1$ with $H = 2$ in Example 3.1, estimate its density via kernel density estimation, and compute the KL divergence to its theoretical limit $Z^{y^1}$, the density of which is also approximated via Monte Carlo sampling. Figure 1(a) plots the estimated densities of our first trial, showing that the density of $y_1^1$ converges rapidly to that of $Z^{y^1}$ as $n$ increases. Figure 1(b) quantifies this convergence by plotting the average log-KL divergence against $\log_4(n)$, with error bars showing one standard deviation across the 10 trials, demonstrating a consistent decay with growing width.

## 5.2 Scalings of the Dot-product Score and Finiteness of Heads

We next investigate two facets of finite-width behavior in Example 3.1, which are the effect of different scaling rules on the variability of the dot-product score and the impact of a finite number of heads on the attention output. For comparison, [Yan19b] assumes $1/n$-scaling, and the infinite-head result by [HBSDN20] applies only in the limit $n, H \to \infty$.

Figure 2(a) shows the histogram of the empirical distribution of the dot-product score $p_{1,1}^{(1)}$ at width $n = 256$ under the two scaling schemes $1/\sqrt{n}$ and $1/n$, together with the infinite-width limit distribution $\mathring{p}_{1,1}^1$. Even at this moderate width, the $1/n$-scaled scores are tightly concentrated around zero, whereas the $1/\sqrt{n}$-scaled scores remain spread out in a nondegenerate fashion, confirming that only the latter preserves variability away from zero at large $n$.

Figure 2(b) reports histograms of the attention output $y_1^1$ for $n = 256$ with head counts $H = 1$ and $H = 256$, overlaid with the infinite-width densities of $Z^{y^1}$, which are approximated via Monte Carlo sampling. The close agreement between finite-width histograms and their infinite-width limit curves, even when $H$ grows on the same order as $n$, demonstrates the robustness of our infinite-width approximation in the growing number of heads. Furthermore, it is observed that as $H$ increases, our non-Gaussian distribution approaches the Gaussian distribution from [HBSDN20].

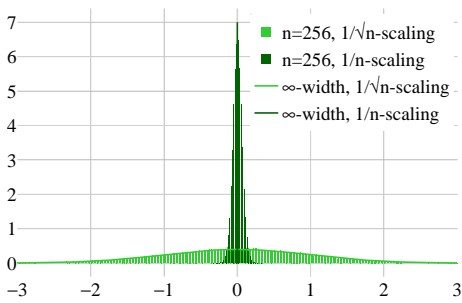
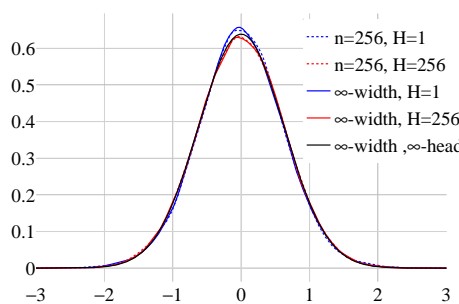

(a) Dot-product score under different scalings.

(b) Varying head count of the attention output.

Figure 2: Visualization of the dot-product score $p_{1,1}^{(1)}$ and attention output $y_1^1$, as defined in Example 3.1, comparing finite-width behavior to their infinite-width limits. **(a)** Histogram of the empirical distribution of $p_{1,1}^{(1)}$ for $n = 256$ alongside the plot of its infinite-width limit distribution $\mathring{p}_{1,1}^1$, under two scaling schemes; $1/\sqrt{n}$ and $1/n$. The $1/n$-scaled score collapses to zero in the infinite-width limit, while the $1/\sqrt{n}$-scaled score retains a nondegenerate distribution. **(b)** Kernel density estimates of the empirical distribution of $y_1^1$ for $n = 256$ (dashed lines) alongside the plot of its infinite-width limit distribution $Z^{y^1}$ (solid lines), varying head counts $H \in \{1, 256\}$. The black solid line represents the density of the infinite-head limit distribution from [HBSDN20]. This demonstrates that our theoretical prediction remains accurate even when $H$ grows, and it approaches the infinite-head limit.

Beyond the choice of scaling, our work provides a general perspective on the number of attention heads: our theory provides an accurate characterization for both finite and large numbers of heads, effectively subsuming the large-$H$ regime within a single framework. This robustness is confirmed by our experiments, which show our theoretical predictions remain highly accurate as $H$ grows large (Figure 2(b)), as well as in low-rank attention setting where the number of heads $H$ increases proportionally with the network width $n$ (Appendix B.2).

## 6 Conclusion

In this paper, we rigorously analyzed the infinite-width limit distribution of outputs from a single attention layer using the Tensor Programs framework. Specifically, we theoretically showed and empirically confirmed that the attention outputs converge to a non-Gaussian distribution under realistic conditions with finite heads and standard $1/\sqrt{n}$-scaling.

Looking forward, we believe our framework can serve as a foundation for future extensions to deep Transformer architectures. We predict that when layers are stacked as in Transformers, not only attention layers but also MLP layers converge to non-Gaussian distributions in the infinite-width limit. The non-Gaussianity of attention layers (and possibly MLP layers) is not merely a theoretical curiosity, but it has profound implications for the learning dynamics of attention-based models. For instance, the presence of higher-order moments associated with such non-Gaussianity could influence signal propagation by preserving feature variability across layers, thereby reducing the risk of rank collapse. Furthermore, the anisotropy induced by non-Gaussianity may lead to more irregular curvature in the optimization landscape, possibly affecting the convergence properties of training dynamics (e.g., through sharper gradients or more prominent saddle regions). Consequently, we argue that a new framework distinct from existing Gaussian-based approaches is essential for analyzing deep architectures. Our work provides an important first step in this direction.

As a limitation of this study, our analysis assumes a constant sequence length and batch size, following the Tensor Programs framework. Extending the framework to handle growing sequence lengths is an important direction for future research.

## Acknowledgements

We are grateful to the reviewers for their insightful comments. We also thank Yuta Matano for helpful discussions and for pointing out the sub-Gaussianity of the limiting distribution of the attention

output. Mana Sakai was supported by RIKEN Junior Research Associate Program. Ryo Karakida was supported by JST FOREST (Grant No. JPMJFR226Q) and JSPS KAKENHI (Grant No. 22H05116). Masaaki Imaizumi was supported by JSPS KAKENHI (Grant No. 24K02904), JST CREST (Grant No. JPMJCR21D2), and JST FOREST (Grant No. JPMJFR216I).

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

# A    Notation Summary

This section summarizes the key notations. We adopt two main conventions for simplicity. First, for finite-width vectors and matrices, superscripts serve as indices to distinguish variables (e.g., $x^i \in \mathbb{R}^n$, $W^i \in \mathbb{R}^{n \times n}$), while subscripts denote their components (e.g., $x^i_\alpha$, $W^i_{\alpha\beta} \in \mathbb{R}$). Second, the dependence on the network width $n$ for all finite-width variables is kept implicit (e.g., we write $x^i$ instead of $x^i(n)$). For other notations, please refer to Table 1.

Table 1: General Notations

| Symbol | Description |
|---|---|
| $n$ | The dimensionality of vector spaces, corresponding to the network's width. |
| $H$ | The number of heads in the multi-head attention mechanism. |
| $s$ | The spatial dimension of input sequences (i.e., sequence length). |
| $W$ | A generic weight matrix in $\mathbb{R}^{n \times n}$ with elements $W_{\alpha\beta} \sim \mathcal{N}(0, \sigma_W^2/n)$. |
| $Z^h$ | A random variable for the infinite-width limit of a vector $h$. |
| $\mathring{p}$ | A random variable for the infinite-width limit of a scalar dot-product $p$. |
| $\boldsymbol{x}_{1:k}$ | The vector $(x_1, \ldots, x_k)$. |
| $\boldsymbol{x}_\alpha^{1:k}$ | The vector $(x_\alpha^1, \ldots, x_\alpha^k)$. |

# B    Simulation Details and Supplementary Analysis

## B.1    General Experimental Setup

Unless otherwise noted, the simulations presented in this paper set the spatial dimension to $s = 4$. The core experimental setup follows that described in Example 3.1, which is outlined in Algorithm 1.

---

**Algorithm 1** Multi-Head Attention (Example 3.1)

---

**Input:** $\{x^i\}_{i \in [s]}$    ▷ $\mathbb{R}^n$ input vectors for a sequence of length $s$
**Input:** $\{W^{Q,a}, W^{K,a}, W^{V,a}, W^{O,a}\}_{a \in [H]}$    ▷ $\mathbb{R}^{n \times n}$ weight matrices for $H$ heads

> **for** $a \in [H]$ **do**
>> **for** $i \in [s]$ **do**
>>> $q^{a,i} \leftarrow W^{Q,a} x^i$    ▷ MatMul: Query vectors
>>> $k^{a,i} \leftarrow W^{K,a} x^i$    ▷ MatMul: Key vectors
>>> $v^{a,i} \leftarrow W^{V,a} x^i$    ▷ MatMul: Value vectors
>>> $\tilde{v}^{a,i} \leftarrow W^{O,a} v^{a,i}$    ▷ MatMul: Output-projected value vectors
>> **end for**
> **end for**

> **for** $a \in [H]$ **do**
>> **for** $i \in [s]$ **do**
>>> **for** $j \in [s]$ **do**
>>>> $p_{i,j}^{(a)} \leftarrow (q^{a,i})^\top k^{a,j} / \sqrt{n}$    ▷ Scaled dot-product scores
>>> **end for**
>> **end for**
> **end for**

> **for** $i \in [s]$ **do**
>> $y^i \leftarrow \frac{1}{\sqrt{H}} \sum_{a=1}^H \sum_{j=1}^s \text{SoftMax}_j(p_{i,1}^{(a)}, \ldots, p_{i,s}^{(a)}) \tilde{v}^{a,j}$    ▷ Attention output vectors
> **end for**

**Output:** $\{y^i\}_{i \in [s]}$    ▷ $\mathbb{R}^n$ output vectors

---

Specifically, let $x^1, \ldots, x^s$ be outputs of Nonlin. These are obtained by applying a clipping activation function $\psi$ to preceding vectors $h^1, \ldots, h^s \in \mathbb{R}^n$:

$$x_\alpha^i = \psi(h_\alpha^i) = -C1\{h_\alpha^i < -C\} + h_\alpha^i 1\{-C \leq h_\alpha^i \leq C\} + C1\{h_\alpha^i > C\} \quad (\alpha \in [n],\ i \in [s]), \quad (4)$$

where $1\{\cdot\}$ denotes the indicator function and $C$ is a positive constant. The vectors $h^1, \ldots, h^s$ are outputs of MatMul, defined by

$$h^i = W^i h \quad (i \in [s]).$$

Each element of the initial vector $h \in \mathbb{R}^n$ is sampled independently from a standard normal distribution.

For all weight matrices involved in the attention mechanism $W^{Q,a}, W^{K,a}, W^{V,a}, W^{O,a}$ and the matrices $W^i$ generating $x^i$, we set

$$\sigma_{W^{Q,a}}^2 = \sigma_{W^{K,a}}^2 = \sigma_{W^{V,a}}^2 = \sigma_{W^{O,a}}^2 = \sigma_{W^i}^2 = 1.$$

The elements of these weight matrices are independently sampled from $\mathcal{N}(0, \sigma_W^2/n)$, where $\sigma_W^2$ is the respective variance (here, 1).

Under this setup, the input vectors $x^j$ to the attention layer are designed such that their infinite-width limits $Z^{x^j}$ are uncorrelated for $j \neq j'$, i.e.,

$$\mathbb{E}[Z^{x^j} Z^{x^{j'}}] = 0 \quad (j, j' \in [s],\ j \neq j').$$

Furthermore, the infinite-width limit $Z^{h^i}$ is a standard normal random variable, leading to

$$\mathbb{E}[(Z^{x^i})^2] = \mathbb{E}[(\psi(Z^{h^i}))^2] = 2C^2(1 - \Phi(C)) - 2C\phi(C) + 2\Phi(C) - 1,$$

where $C$ is the constant used in the clipping activation, and $\Phi$ and $\phi$ are the cumulative distribution function and probability density function of the standard normal distribution, respectively. As $C \to \infty$, this value converges to 1. In our experiments, we set $C = 100$, and we approximate $\mathbb{E}[(Z^{x^i})^2]$ by 1. Consequently, the covariances of the limiting variables for the vectors $\tilde{v}^{a,j}$ and dot-product scores $\mathring{p}_{i,j}^{(a)}$ simplify as described in Example 3.1,

$$\text{Cov}(Z^{\tilde{v}^{a,j}}, Z^{\tilde{v}^{a',j'}}) = \begin{cases} \mathbb{E}[(Z^{x^j})^2] \approx 1 & (a = a',\ j = j'), \\ 0 & \text{(otherwise)}, \end{cases}$$

and

$$\text{Cov}(\mathring{p}_{i,j}^{(a)}, \mathring{p}_{i',j'}^{(a')}) = \begin{cases} \left(\mathbb{E}[(Z^{x^i})^2]\right)^2 \approx 1 & (a = a',\ i = i',\ j = j'), \\ 0 & \text{(otherwise)}. \end{cases}$$

To estimate the empirical distributions of finite-width attention outputs and their corresponding infinite-width limits, we employ Monte Carlo sampling. For each such estimation, 50,000 samples are drawn, unless otherwise noted. Kernel density estimation (KDE) is used to visualize these empirical distributions.

## B.2 Analysis of Low-Rank Attention

### B.2.1 Specific Setup for Low-Rank Attention

In practice, large-scale Transformers typically assume a specific embedding dimensionality for multi-head self-attention layers. For head counts $H$, the embedding dimension $n$ is set linearly as $n = Hn_H$, where $n_H$ denotes the head dimension and determines the sizes of weight matrices as $W^{Q,a}, W^{K,a}, W^{V,a} \in \mathbb{R}^{n_H \times n}$. Thus, the QK product becomes low-rank relative to the embedding dimension $n$, and the scaling factor is given by $1/\sqrt{n_H}$ as follows:

$$p_{i,j}^{(a)} = \frac{1}{\sqrt{n_H}} (W^{Q,a} x^i)^\top (W^{K,a} x^j) \quad (i, j \in [s],\ a \in [H]).$$

For example, the original Transformer architecture sets $H = 8$ and $n_H = 64$. Large-scale models often increase the number of heads to be on the order of the hidden embedding dimension [EXW+24], as seen in GPT-3 (175B parameters), which sets $H = 96$ and $n_H = 128$.

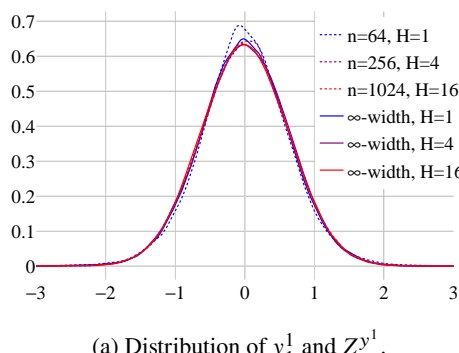 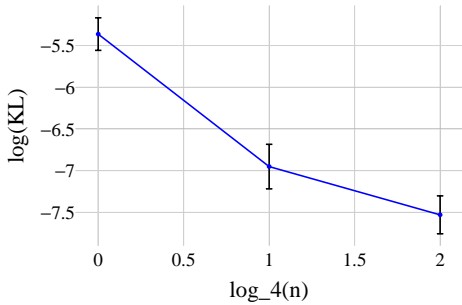

(a) Distribution of $y_1^1$ and $Z^{y^1}$.          (b) KL divergence with error bars.

Figure 3: Comparison of the distribution of the attention output $y_1^1$ and its infinite-width limit $Z^y$ under the low-rank setting. **(a)** Kernel density estimates of the empirical distribution (via Monte Carlo sampling) of $y_1^1$ for various widths $n$ and head counts $H$ (with $n_H = n/H = 64$ is fixed, dashed lines) alongside that of $Z^y$ (solid lines). **(b)** Average of the log-KL divergence $\log \mathrm{KL}(\mathrm{Dist}(y_1^1)\|\mathrm{Dist}(Z^{y^1}))$ over 10 independent trials, plotted against $\log_4(n)$ with error bars indicating one standard deviation.

Additionally, since the output from each head is also of dimension $n_H$ through the value matrix, an output weight $W^{O,a} \in \mathbb{R}^{n \times n_H}$ is applied to map it back to the $n$-dimensional input for the subsequent layer:

$$y^i = \frac{1}{\sqrt{H}} \sum_{a=1}^{H} \sum_{j=1}^{s} \mathrm{SoftMax}_j(p_{i,1}^{(a)}, \ldots, p_{i,s}^{(a)}) W^{O,a} W^{V,a} x^j \quad (i \in [s]).$$

Note that, to ensure the dot-product scores and the attention outputs are of order 1, the weight matrices are randomly initialized with the following scales:

$$W_{\alpha\beta}^{Q,a}, W_{\alpha\beta}^{K,a}, W_{\alpha\beta}^{V,a} \sim N(0, \sigma_W^2/n), \qquad W_{\alpha\beta}^{O,a} \sim N(0, \sigma_W^2/n_H) \quad (\alpha, \beta \in [n]).$$

For simplicity, we set $\sigma_W = 1$.

### B.2.2 Results and Discussion

Finally, we investigate the behavior of the attention output $y^1$ in the low-rank regime described above, where the number of heads $H$ increases proportionally with $n$. Fixing the head dimension $n_H = n/H = 64$, we perform 10 independent experiments for each $(n, H) \in \{(64, 1), (256, 4), (1024, 16)\}$. Figure 3(a) shows the estimated densities of our first trial, and Figure 3(b) plots the average log-KL divergence between the distribution of $y_1^1$ and the corresponding infinite-width limit distribution, with error bars showing one standard deviation across the 10 trials. These figures show the convergence to the infinite-width limit as $n$ and $H$ increase proportionally.

Notably, even in these practically relevant settings employing low-rank attention (where head-specific projections are $n_H \times n$ or $n \times n_H$ rather than the $n \times n$ matrices primarily assumed in our formal derivations in Theorem 3.1), our infinite-width framework continues to provide an excellent approximation. This agreement suggests that the core principles of convergence captured by our theory extend robustly to common architectural variants like low-rank attention (with appropriate scaling considerations), underscoring the practical utility of our theoretical predictions under these structural assumptions common in modern attention models.

### B.3 Additional Experiments on Robustness

To further validate the robustness of our theoretical predictions, we conducted additional experiments by varying key hyperparameters. These experiments investigate the impact of the spatial dimension (i.e., the number of tokens) and the choice of activation function.

### B.3.1 Varying the Spatial Dimension

Our main experiments in Section 5 were conducted with a spatial dimension of $s = 4$. Here, we present the results for an increased spatial dimension of $s = 8$. All other experimental settings remain

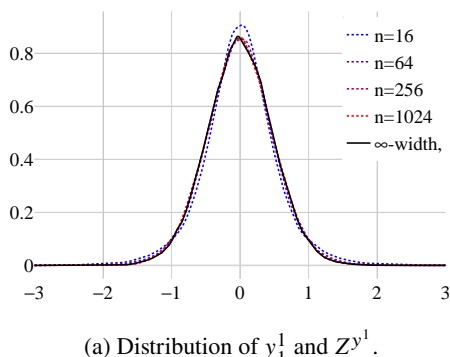
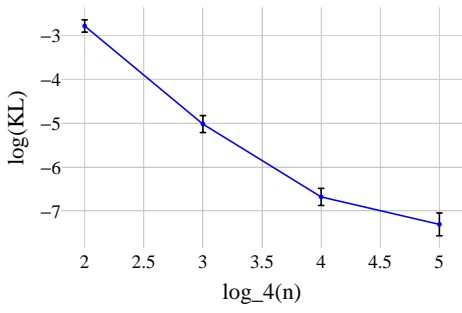

(a) Distribution of $y_1^1$ and $Z^{y^1}$.

(b) KL divergence with error bars.

Figure 4: Comparison of the distribution of the attention output $y_1^1$ and its infinite-width limit $Z^{y^1}$ when $s = 8$. **(a)** Kernel density estimates of the empirical distribution (via Monte Carlo sampling) of $y_1^1$ for widths $n \in \{16, 64, 256, 1024\}$ (dashed lines) alongside that of $Z^{y^1}$ (solid line). **(b)** Average of the log-KL divergence $\log \mathrm{KL}(\mathrm{Dist}(y_1^1)\|\mathrm{Dist}(Z^{y^1}))$ over 10 independent trials, plotted against $\log_4(n)$ with error bars indicating one standard deviation.

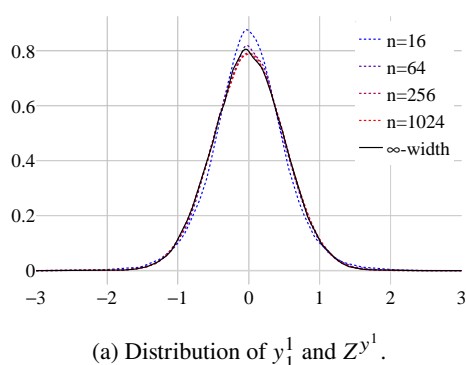
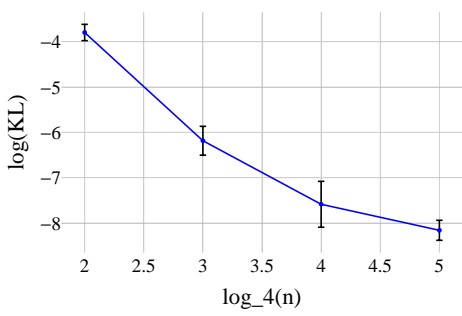

(a) Distribution of $y_1^1$ and $Z^{y^1}$.

(b) KL divergence with error bars.

Figure 5: Comparison of the distribution of the attention output $y_1^1$ and its infinite-width limit $Z^y$ with ReLU activation function. **(a)** Kernel density estimates of the empirical distribution (via Monte Carlo sampling) of $y_1^1$ for various widths $n$ and head counts $H$ (with $n_H = n/H = 64$ is fixed, dashed lines) alongside that of $Z^y$ (solid lines). **(b)** Average of the log-KL divergence $\log \mathrm{KL}(\mathrm{Dist}(y_1^1)\|\mathrm{Dist}(Z^{y^1}))$ over 10 independent trials, plotted against $\log_4(n)$ with error bars indicating one standard deviation.

identical to those described in Appendix B.1. The results, shown in Figure 4, demonstrates that our theory remains accurate even when the number of tokens is changed. This suggests that the convergence to the theoretical limit is robust to changes in the sequence length.

### B.3.2 Varying the Activation Function

The experiments in the main text utilize a clipping activation function. To ensure our findings are not specific to this choice, we also perform experiments with the ReLU activation function, i.e., $\psi(h_\alpha^i) = h_\alpha^i \vee 0$ in Eq. (4), which is widely used in modern neural networks.[5] The covariances of the limiting variables for the vectors $\tilde{v}^{a,j}$ and the dot-product scores $\mathring{p}_{i,j}^{(a)}$ are

$$\mathrm{Cov}(Z^{\tilde{v}^{a,j}}, Z^{\tilde{v}^{a',j'}}) = 1\{a = a'\} \left[ \frac{1}{2\pi} + \left(\frac{1}{2} - \frac{1}{2\pi}\right) 1\{j = j'\} \right]$$

and

$$\mathrm{Cov}(\mathring{p}_{i,j}^{(a)}, \mathring{p}_{i',j'}^{(a')}) = 1\{a = a'\} \left[ \frac{1}{2\pi} + \left(\frac{1}{2} - \frac{1}{2\pi}\right) 1\{i = i'\} \right] \left[ \frac{1}{2\pi} + \left(\frac{1}{2} - \frac{1}{2\pi}\right) 1\{j = j'\} \right].$$

---

[5]We note that the ReLU function is not bounded and consequently, our theory does not directly apply. However, as mentioned in the main text, the boundedness assumption is not essential to our theory.

For each experiment, we draw 100,000 samples for both the finite-width attention output and its corresponding infinite-width limit, an increase from 50,000. The experimental setup is otherwise identical to that described in Appendix B.1.

Figure 5 confirms a strong agreement between the empirical distributions and our theoretical predictions. This indicates that our theory is robust to this change in activation function, strengthening its applicability to a broader range of practical model architectures.

## C  Mathematical Tools

### C.1  Basics

**Lemma C.1.** *For $1 \le m \le \infty$ and $a_1, \ldots, a_k \in \mathbb{R}$, we have*

$$\left| \sum_{i=1}^{k} a_i \right|^m \le \left( \sum_{i=1}^{k} |a_i| \right)^m \le k^{m-1} \sum_{i=1}^{k} |a_i|^m.$$

*Proof.* The first inequality is an application of the triangle inequality. The second inequality follows from Jensen's inequality. Since $m \ge 1$, the function $x \mapsto x^m$ on $[0, \infty)$ is convex, and thus Jensen's inequality implies

$$\left( \sum_{i=1}^{k} |a_i| \right)^m = k^m \left( \frac{1}{k} \sum_{i=1}^{k} |a_i| \right)^m \le k^m \frac{1}{k} \sum_{i=1}^{k} |a_i|^m = k^{m-1} \sum_{i=1}^{k} |a_i|^m$$

as desired. □

**Lemma C.2** (Portmanteau lemma (Lemma 2.2 in [Vaa98]))**.** *The following conditions are equivalent.*

(i) $X_n \xrightarrow{d} X$.

(ii) $\mathbb{E}[f(X_n)] \to \mathbb{E}[f(X)]$ *for all bounded and continuous function $f$.*

(iii) $\mathbb{E}[f(X_n)] \to \mathbb{E}[f(X)]$ *for all bounded and Lipschitz function $f$.*

(iv) $\mathrm{P}(X_n \in B) \to \mathrm{P}(X \in B)$ *for all Borel sets $B$ with $\mathrm{P}(X \in \delta B) = 0$, where $\delta B$ denotes the boundary of $B$.*

**Fact C.1.** *Suppose $\{X_n\}_{n \in \mathbb{N}}$ is a sequence of integrable random variables that converges in probability to $X$. Then the following statements are equivalent.*

(i) *The sequence $\{X_n\}_{n \in \mathbb{N}}$ is uniformly integrable.*

(ii) $\mathbb{E}(|X_n|) \to \mathbb{E}(|X|) < \infty$.

**Remark C.1.** If $X_n$ converges to 0 in probability, then by Fact C.1, we have

$$\mathbb{E}(|X_n|) = o(1) \quad \Longleftrightarrow \quad \{X_n\}_{n \in \mathbb{N}} \text{ is uniformly integrable.}$$

Moreover, since $|\mathbb{E}(X_n)| \le \mathbb{E}(|X_n|)$, it follows that $\mathbb{E}(X_n) = o(1)$.

**Fact C.2.** *Suppose there exists $\delta > 1$ such that $\sup_n \mathbb{E}(|X_n|^\delta) < \infty$. Then the sequence $\{X_n\}_{n \in \mathbb{N}}$ is uniformly integrable.*

### C.2  Pseudo-Lipschitz Functions

**Definition C.1** (Pseudo-Lipschitz functions [BM11])**.** Let $d > 1$ be a constant. A function $f : \mathbb{R}^k \to \mathbb{R}$ is pseudo-Lipschitz of order $d$ if there exists a constant $C > 0$ such that, for all $x, y \in \mathbb{R}^k$,

$$|f(x) - f(y)| \le C\|x - y\|(1 + \|x\|^{d-1} + \|y\|^{d-1})$$

holds.

**Fact C.3.** *The following statements hold.*

(i) *A Lipschitz function is pseudo-Lipschitz of order d for all $d > 1$.*

*(ii) A pseudo-Lipschitz function (of any given order) is continuous.*

In this paper, we refer to a function as pseudo-Lipschitz if it is pseudo-Lipschitz of order $d$ for some $d \in [2, \infty)$.

**Proposition C.3.** *Suppose $f : \mathbb{R}^k \to \mathbb{R}$ and $g_i : \mathbb{R}^\ell \to \mathbb{R}$ ($i \in [k]$) are pseudo-Lipschitz. Then the function $h : \mathbb{R}^\ell \to \mathbb{R}$ defined by $h(x) = f(g_1(x), \dots, g_k(x))$ is also pseudo-Lipschitz.*

*Proof.* Suppose $f$ is pseudo-Lipschitz of order $d_0 + 1$ and each $g_i$ is pseudo-Lipschitz of order $d_i + 1$. Define a function $g : \mathbb{R}^\ell \to \mathbb{R}^k$ and a constant $d \geq 1$ by

$$g(x) = (g_1(x), \dots, g_k(x)), \quad d = \max\{d_1, \dots, d_k\}.$$

Applying the pseudo-Lipschitz bounds for $f$ and each $g_i$ gives

$$|h(x) - h(x')| \lesssim \|g(x) - g(x')\| \left(1 + \|g(x)\|^{d_0} + \|g(x')\|^{d_0}\right)$$

and

$$|g_i(x) - g_i(x')| \lesssim \|x - x'\| \left(1 + \|x\|^{d_i} + \|x'\|^{d_i}\right) \lesssim \|x - x'\| \left(1 + \|x\|^{d} + \|x'\|^{d}\right).$$

The last inequality implies

$$\|g(x) - g(x')\| = \left(\sum_{i=1}^{k} |g_i(x) - g_i(x')|^2\right)^{1/2} \lesssim \|x - x'\| \left(1 + \|x\|^{d} + \|x'\|^{d}\right).$$

On the other hand, since

$$\|g(x)\|^2 = \sum_{i=1}^{k} |g_i(x)|^2 \leq \left(\sum_{i=1}^{k} |g_i(x)|\right)^2$$

holds in general, Lemma C.1 implies

$$\|g(x)\|^{d_0} \leq \left(\sum_{i=1}^{k} |g_i(x)|\right)^{d_0} \lesssim \sum_{i=1}^{k} |g_i(x)|^{d_0}.$$

The pseudo-Lipschitz property of each $g_i$ yields

$$|g_i(x)| \leq |g_i(0)| + |g_i(x) - g_i(0)| \lesssim 1 + \|x\|(1 + \|x\|^{d_i}) \lesssim 1 + \|x\|^{d_i+1} \lesssim 1 + \|x\|^{d+1}.$$

Hence, by Lemma C.1, we have

$$\|g(x)\|^{d_0} \lesssim (1 + \|x\|^{d+1})^{d_0} \lesssim 1 + \|x\|^{d_0(d+1)}.$$

Combining these elements gives

$$|h(x) - h(x')| \lesssim \|x - x'\| \left(1 + \|x\|^{d} + \|x'\|^{d}\right) \left(1 + \|x\|^{d_0(d+1)} + \|x'\|^{d_0(d+1)}\right).$$

We expand the product as

$$(1 + \|x\|^{d} + \|x'\|^{d})(1 + \|x\|^{d_0(d+1)} + \|x'\|^{d_0(d+1)})$$
$$= 1 + \|x\|^{d} + \|x'\|^{d} + \|x\|^{d_0(d+1)} + \|x'\|^{d_0(d+1)} + \|x\|^{d+d_0(d+1)} + \|x'\|^{d+d_0(d+1)}$$
$$+ \|x\|^{d}\|x'\|^{d_0(d+1)} + \|x'\|^{d}\|x\|^{d_0(d+1)}.$$

Observe that each of the first seven terms are bounded by $1 + \|x\|^{a} + \|x'\|^{a}$, where $a$ is given by

$$a = d + d_0(d + 1).$$

For the remaining two terms, we apply the weighted AM-GM inequality to obtain

$$\|x\|^{d}\|x'\|^{d_0(d+1)} = (\|x\|^{a})^{\frac{d}{a}} (\|x'\|^{a})^{\frac{d_0(d+1)}{a}} \leq \frac{d}{a}\|x\|^{a} + \frac{d_0(d+1)}{a}\|x'\|^{a} \leq \|x\|^{a} + \|x'\|^{a}.$$

The same bound applies to $\|x'\|^{d}\|x\|^{d_0(d+1)}$. Therefore the entire product satisfies

$$(1 + \|x\|^{d} + \|x'\|^{d})(1 + \|x\|^{d_0(d+1)} + \|x'\|^{d_0(d+1)}) \lesssim 1 + \|x\|^{a} + \|x'\|^{a},$$

which implies that $h$ is pseudo-Lipschitz of order $a$. $\qquad\square$

**Lemma C.4.** *Define $f : \mathbb{R}^2 \to \mathbb{R}$ by $f(x, y) = xy$. Then $f$ is pseudo-Lipschitz of order $d$ for every $d \in [2, \infty)$.*

*Proof.* For $(x, y), (x', y') \in \mathbb{R}^2$, we have

$$|f(x, y) - f(x', y')| = |xy - x'y'| = |x(y - y') + (x - x')y'|$$
$$\leq |x||y - y'| + |x - x'||y'| \leq \|(x, y) - (x', y')\|(|x| + |y'|).$$

Observe that for any $d \geq 2$, we have

$$|x| \leq \|(x, y)\| \leq 1 + \|(x, y)\|^{d-1}, \quad |y'| \leq \|(x', y')\| \leq 1 + \|(x', y')\|^{d-1},$$

and consequently, we have

$$|x| + |y'| \lesssim 1 + \|(x, y)\|^{d-1} + \|(x', y')\|^{d-1}.$$

This gives us

$$|f(x, y) - f(x', y')| \lesssim \|(x, y) - (x', y')\|(1 + \|(x, y)\|^{d-1} + \|(x', y')\|^{d-1}),$$

which shows that $f$ is pseudo-Lipschitz of order $d$. $\qquad\square$

## D  Remaining proofs

### D.1  Proof of Theorem 4.1

In this section we provide detailed proofs for the results sketched in Section 4, thereby completing the proof of Theorem 4.1.

Throughout, $\mathbb{E}[\cdot \mid X]$ denotes the conditional expectation with respect to the $\sigma$-algebra $\sigma(X)$. Since conditional expectations are only defined up to almost-sure equality, we omit "a.s." when writing "$\overset{\text{a.s.}}{=}$" in this context.

#### D.1.1  Weak Convergence of the Dot Products

In this section we prove the following proposition.

**Proposition D.1.** *Under the assumptions of Theorem 4.1, the vector $(p_1, \ldots, p_r)$ converges in distribution to $(\mathring{p}_1, \ldots, \mathring{p}_r)$, which is the Gaussian vector defined in Definition 3.1.*

The proof of Proposition D.1 relies on the next lemma, which is an application of Theorem 2 in [BCRT58].

**Lemma D.2.** *For each $n \in \mathbb{N}$, let $\{X_\alpha\}_{\alpha \in [n]}$ be an exchangeable sequence of random variables satisfying*

$$\mathbb{E}[X_\alpha] = 0, \quad \mathbb{E}[X_\alpha^2] = \sigma_n^2, \quad \sigma_n^2 \to \sigma_*^2 \geq 0 \quad (n \to \infty).$$

*Set $S = \sum_{\alpha=1}^n X_\alpha / \sqrt{n}$. Assume the following conditions:*

*(a) $\mathbb{E}(X_1 X_2) = o(1/n)$.*

*(b) $\lim_{n \to \infty} \mathbb{E}(X_1^2 X_2^2) = \sigma_*^4$.*

*(c) $\mathbb{E}(|X_1|^3) = o(\sqrt{n})$.*

*Then, $S$ converges in distribution to $Z$, where the random variable $Z$ satisfies*

$$Z \overset{\text{a.s.}}{=} 0 \quad \text{if} \quad \sigma_*^2 = 0, \qquad Z \sim N(0, \sigma_*^2) \quad \text{otherwise.}$$

We introduce the following notation. For any two matrices $W^{i,j}$ and $W^{i',j'}$ appearing in the program, define $d_{(i,j)}^{(i',j')}$ by

$$d_{(i,j)}^{(i',j')} = \begin{cases} 1 & (\text{if } W^{i,j} \text{ and } W^{i',j'} \text{ are the same matrices}), \\ 0 & (\text{otherwise}). \end{cases}$$

It is important to note that $d^{(i',j')}_{(i,j)}$ is always a deterministic value that is independent of $n$, and is fixed by the program architecture. According to the sampling rule explained in Section 2.1, the matrices $W^{i,j}$ and $W^{i'j'}$ are sampled independently whenever $d^{(i',j')}_{(i,j)}$ is zero. In particular, Assumption 3.1 gives

$$d^{(i,2)}_{(i,1)} = 0 \quad (i \in [r]). \tag{5}$$

Let $t_1, \ldots, t_r$ be arbitrary constants. We define

$$S = \sum_{i=1}^{r} t_i p_i = \sum_{i=1}^{r} t_i \left( \frac{1}{\sqrt{n}} \sum_{\alpha=1}^{n} \sum_{\gamma_1=1}^{n} \sum_{\gamma_2=1}^{n} W^{i,1}_{\alpha\gamma_1} W^{i,2}_{\alpha\gamma_2} x^{i,1}_{\gamma_1} x^{i,2}_{\gamma_2} \right) = \frac{1}{\sqrt{n}} \sum_{\alpha=1}^{n} X_\alpha,$$

where $X_\alpha$ is given by

$$X_\alpha = \sum_{i=1}^{r} \sum_{\gamma_1, \gamma_2=1}^{n} t_i W^{i,1}_{\alpha\gamma_1} W^{i,2}_{\alpha\gamma_2} x^{i,1}_{\gamma_1} x^{i,2}_{\gamma_2}.$$

Lemmas D.3–D.8 show that the sequence $\{X_\alpha\}_{\alpha\in[n]}$ satisfies the conditions of Lemma D.2. The proof of Proposition D.1 is completed by applying Lemma D.2 to $S = \sum_{\alpha=1}^{n} X_\alpha/\sqrt{n}$ and then invoking the Cramér–Wold device.

**Lemma D.3.** *The sequence $\{X_\alpha\}_{\alpha\in[n]}$ is exchangeable.*

*Proof.* By the sampling rule, an element $W^{i,j}_{\alpha\beta}$ of the random matrix $W^{i,j}$ independently and identically follows $\mathcal{N}(0, \sigma^2_{W^{i,j}}/n)$ for $\alpha, \beta \in [n]$. Hence, conditional on $\{x^{i,j} : i \in [r], \ j \in [2]\}$, the random variables $X_1, \ldots, X_n$ are i.i.d. Hence, by de Finetti's theorem, $\{X_\alpha\}_{\alpha\in[n]}$ is exchangeable. $\qquad\square$

**Lemma D.4.** $\mathbb{E}(X_\alpha) = 0$ *holds for every $\alpha \in [n]$.*

*Proof.* We compute $\mathbb{E}(X_\alpha) = \sum_{i=1}^{r} \sum_{\gamma_1,\gamma_2=1}^{n} t_i \mathbb{E}\left(W^{i,1}_{\alpha\gamma_1}\right) \mathbb{E}\left(W^{i,2}_{\alpha\gamma_2}\right) \mathbb{E}\left(x^{i,1}_{\gamma_1} x^{i,2}_{\gamma_2}\right) = 0.$ $\qquad\square$

**Lemma D.5.** *We have $\lim_{n\to\infty} \mathbb{E}(X^2_\alpha) = \sigma^2_*$ with*

$$\sigma^2_* = \sum_{i_1, i_2=1}^{r} t_{i_1} t_{i_2} \mathbb{E}\left[ Z^{g^{i_1,1}} Z^{g^{i_1,2}} Z^{g^{i_2,1}} Z^{g^{i_2,2}} \right]$$

$$= \sum_{i_1, i_2=1}^{r} \sum_{(j,j')\in J} t_{i_1} t_{i_2} \sigma^2_{W^{i_1,1}} \sigma^2_{W^{i_1,2}} \mathbb{E}\left[ Z^{x^{i_1,1}} Z^{x^{i_2,j}} \right] \mathbb{E}\left[ Z^{x^{i_1,2}} Z^{x^{i_2,j'}} \right] d^{(i_2,j)}_{(i_1,1)} d^{(i_2,j')}_{(i_1,2)},$$

*where we defined $J = \{(1,2), (2,1)\}$.*

*Proof.* For any $\alpha \in [n]$, we have
$\mathbb{E}(X^2_\alpha)$

$$= \mathbb{E}\left[ \left( \sum_{i_1=1}^{r} \sum_{\gamma_1,\gamma_2=1}^{n} t_{i_1} W^{i_1,1}_{\alpha\gamma_1} W^{i_1,2}_{\alpha\gamma_2} x^{i_1,1}_{\gamma_1} x^{i_1,2}_{\gamma_2} \right) \left( \sum_{i_2=1}^{r} \sum_{\gamma_3,\gamma_4=1}^{n} t_{i_2} W^{i_2,1}_{\alpha\gamma_3} W^{i_2,2}_{\alpha\gamma_4} x^{i_2,1}_{\gamma_3} x^{i_2,2}_{\gamma_4} \right) \right]$$

$$= \sum_{i_1,i_2=1}^{r} \sum_{\gamma_1,\ldots,\gamma_4=1}^{n} t_{i_1} t_{i_2} \mathbb{E}\left( W^{i_1,1}_{\alpha\gamma_1} W^{i_1,2}_{\alpha\gamma_2} W^{i_2,1}_{\alpha\gamma_3} W^{i_2,2}_{\alpha\gamma_4} \right) \mathbb{E}\left( x^{i_1,1}_{\gamma_1} x^{i_1,2}_{\gamma_2} x^{i_2,1}_{\gamma_3} x^{i_2,2}_{\gamma_4} \right)$$

$$= \sum_{i_1,i_2=1}^{r} \sum_{\gamma_1,\gamma_2=1}^{n} \sum_{(j,j')\in J} t_{i_1} t_{i_2} \mathbb{E}\left[ (W^{i_1,1}_{\alpha\gamma_1})^2 \right] \mathbb{E}\left[ (W^{i_1,2}_{\alpha\gamma_2})^2 \right] \mathbb{E}\left( x^{i_1,1}_{\gamma_1} x^{i_2,j}_{\gamma_1} x^{i_1,2}_{\gamma_2} x^{i_2,j'}_{\gamma_2} \right) d^{(i_2,j)}_{(i_1,1)} d^{(i_2,j')}_{(i_1,2)}$$

$$= \sum_{i_1,i_2=1}^{r} \sum_{(j,j')\in J} t_{i_1} t_{i_2} \sigma^2_{W^{i_1,1}} \sigma^2_{W^{i_1,2}} \mathbb{E}\left[ \left( \frac{1}{n} \sum_{\gamma_1=1}^{n} x^{i_1,1}_{\gamma_1} x^{i_2,j}_{\gamma_1} \right) \left( \frac{1}{n} \sum_{\gamma_2=1}^{n} x^{i_1,2}_{\gamma_2} x^{i_2,j'}_{\gamma_2} \right) \right] d^{(i_2,j)}_{(i_1,1)} d^{(i_2,j')}_{(i_1,2)}$$

$$\xrightarrow{n\to\infty} \sum_{i_1,i_2=1}^{r} \sum_{(j,j')\in J} t_{i_1} t_{i_2} \sigma^2_{W^{i_1,1}} \sigma^2_{W^{i_1,2}} \mathbb{E}\left[ Z^{x^{i_1,1}} Z^{x^{i_2,j}} \right] \mathbb{E}\left[ Z^{x^{i_1,2}} Z^{x^{i_2,j'}} \right] d^{(i_2,j)}_{(i_1,1)} d^{(i_2,j')}_{(i_1,2)},$$

where the convergence follows from Lemma D.9. Finally, Eq. (5) and Definition 3.1 imply that

$$\sigma^2_{W^{i_1,1}}\sigma^2_{W^{i_1,2}}\mathbb{E}\left[Z^{x^{i_1,1}}Z^{x^{i_2,j}}\right]\mathbb{E}\left[Z^{x^{i_1,2}}Z^{x^{i_2,j'}}\right]d^{(i_2,j)}_{(i_1,1)}d^{(i_2,j')}_{(i_1,2)} = \mathbb{E}\left[Z^{g^{i_1,1}}Z^{g^{i_1,2}}Z^{g^{i_2,1}}Z^{g^{i_2,2}}\right]$$

holds for any $i_1, i_2 \in [r]$ and $(j, j') \in J$. $\qquad\square$

**Lemma D.6.** $\mathbb{E}(X_\alpha X_\beta) = 0$ *holds for every* $\alpha, \beta \in [n], \alpha \neq \beta$.

*Proof.* For $\alpha \neq \beta$, we have

$$\mathbb{E}(X_\alpha X_\beta) = \mathbb{E}\left[\left(\sum_{i_1=1}^{r}\sum_{\gamma_1,\gamma_2=1}^{n} t_{i_1} W^{i_1,1}_{\alpha\gamma_1} W^{i_1,2}_{\alpha\gamma_2} x^{i_1,1}_{\gamma_1} x^{i_1,2}_{\gamma_2}\right)\left(\sum_{i_2=1}^{r}\sum_{\gamma_3,\gamma_4=1}^{n} t_{i_2} W^{i_2,1}_{\beta\gamma_3} W^{i_2,2}_{\beta\gamma_4} x^{i_2,1}_{\gamma_3} x^{i_2,2}_{\gamma_4}\right)\right]$$

$$= \sum_{i_1,i_2=1}^{r}\sum_{\gamma_1,\ldots,\gamma_4=1}^{n} t_{i_1}t_{i_2}\mathbb{E}\left(W^{i_1,1}_{\alpha\gamma_1}\right)\mathbb{E}\left(W^{i_1,2}_{\alpha\gamma_2}\right)\mathbb{E}\left(W^{i_2,1}_{\beta\gamma_3}\right)\mathbb{E}\left(W^{i_2,2}_{\beta\gamma_4}\right)\mathbb{E}\left(x^{i_1,1}_{\gamma_1}x^{i_1,2}_{\gamma_2}x^{i_2,1}_{\gamma_3}x^{i_2,2}_{\gamma_4}\right)$$

$$= 0$$

as desired. $\qquad\square$

**Lemma D.7.** $\lim_{n\to\infty}\mathbb{E}(X^2_\alpha X^2_\beta) = \sigma^4_*$ *holds for every* $\alpha, \beta \in [n], \alpha \neq \beta$.

*Proof.* By a calculation similar to Lemma D.5, we have

$$\mathbb{E}(X^2_\alpha X^2_\beta)$$

$$= \sum_{i_1,\ldots,i_4=1}^{r}\sum_{(j_1,j'_1),(j_2,j'_2)\in J^2} t_{i_1}t_{i_2}t_{i_3}t_{i_4}\sigma^2_{W^{i_1,1}}\sigma^2_{W^{i_2,2}}\sigma^2_{W^{i_3,1}}\sigma^2_{W^{i_4,2}}d^{(i_2,j_1)}_{(i_1,1)}d^{(i_2,j'_1)}_{(i_1,2)}d^{(i_4,j_2)}_{(i_3,1)}d^{(i_4,j'_2)}_{(i_3,2)}$$

$$\times \mathbb{E}\left[\left(\frac{1}{n}\sum_{\gamma_1=1}^{n} x^{i_1,1}_{\gamma_1}x^{i_2,j_1}_{\gamma_1}\right)\left(\frac{1}{n}\sum_{\gamma_2=1}^{n} x^{i_1,2}_{\gamma_2}x^{i_2,j'_1}_{\gamma_2}\right)\left(\frac{1}{n}\sum_{\gamma_3=1}^{n} x^{i_3,1}_{\gamma_3}x^{i_4,j_2}_{\gamma_3}\right)\left(\frac{1}{n}\sum_{\gamma_4=1}^{n} x^{i_3,2}_{\gamma_4}x^{i_4,j'_2}_{\gamma_4}\right)\right]$$

$$\xrightarrow{n\to\infty} \sum_{i_1,\ldots,i_4=1}^{r}\sum_{(j_1,j'_1),(j_2,j'_2)\in J^2} t_{i_1}t_{i_2}t_{i_3}t_{i_4}\sigma^2_{W^{i_1,1}}\sigma^2_{W^{i_2,2}}\sigma^2_{W^{i_3,1}}\sigma^2_{W^{i_4,2}}d^{(i_2,j_1)}_{(i_1,1)}d^{(i_2,j'_1)}_{(i_1,2)}d^{(i_4,j_2)}_{(i_3,1)}d^{(i_4,j'_2)}_{(i_3,2)}$$

$$\times \mathbb{E}\left[Z^{x^{i_1,1}}Z^{x^{i_2,j_1}}\right]\mathbb{E}\left[Z^{x^{i_1,2}}Z^{x^{i_2,j'_1}}\right]\mathbb{E}\left[Z^{x^{i_3,1}}Z^{x^{i_4,j_2}}\right]\mathbb{E}\left[Z^{x^{i_3,2}}Z^{x^{i_4,j'_2}}\right],$$

where the convergence follows from Lemma D.9. Observe that this limit is equivalent to $\sigma^4_*$. $\qquad\square$

**Lemma D.8.** $\mathbb{E}(|X_\alpha|^3) = o(\sqrt{n})$ *holds as* $n \to \infty$ *for every* $\alpha \in [n]$.

*Proof.* By the Lyapunov inequality, we have

$$\frac{1}{\sqrt{n}}\mathbb{E}(|X_\alpha|^3) \leq \frac{1}{\sqrt{n}}\left(\mathbb{E}(X^4_\alpha)\right)^{\frac{3}{4}} \leq \frac{1}{\sqrt{n}}\left(\sup_n \mathbb{E}(X^4_\alpha)\right)^{\frac{3}{4}}.$$

Thus, it suffices to show that $\sup_n \mathbb{E}(X^4_\alpha) < \infty$ holds. We can express $\mathbb{E}(X^4_\alpha)$ as

$$\mathbb{E}(X^4_\alpha) = \sum_{i_1,\ldots,i_4=1}^{r}\sum_{\gamma_1,\ldots,\gamma_8=1}^{n} t_{i_1}t_{i_2}t_{i_3}t_{i_4}\mathbb{E}\left[W^{i_1,1}_{\alpha\gamma_1}W^{i_2,1}_{\alpha\gamma_2}W^{i_3,1}_{\alpha\gamma_3}W^{i_4,1}_{\alpha\gamma_4}W^{i_1,2}_{\alpha\gamma_5}W^{i_2,2}_{\alpha\gamma_6}W^{i_3,2}_{\alpha\gamma_7}W^{i_4,2}_{\alpha\gamma_8}\right]$$

$$\times \mathbb{E}\left[x^{i_1,1}_{\gamma_1}x^{i_2,1}_{\gamma_2}x^{i_3,1}_{\gamma_3}x^{i_4,1}_{\gamma_4}x^{i_1,2}_{\gamma_5}x^{i_2,2}_{\gamma_6}x^{i_3,2}_{\gamma_7}x^{i_4,2}_{\gamma_8}\right].$$

Applying the Cauchy–Schwarz inequality yields

$$\mathbb{E}\left[x^{i_1,1}_{\gamma_1}x^{i_2,1}_{\gamma_2}x^{i_3,1}_{\gamma_3}x^{i_4,1}_{\gamma_4}x^{i_1,2}_{\gamma_5}x^{i_2,2}_{\gamma_6}x^{i_3,2}_{\gamma_7}x^{i_4,2}_{\gamma_8}\right]$$

$$\leq \left(\prod_{k=1}^{4}\mathbb{E}\left[(x^{i_j,1}_{\gamma_j})^8\right]\right)^{\frac{1}{8}}\left(\prod_{k=1}^{4}\mathbb{E}\left[(x^{i_j,2}_{\gamma_{4+j}})^8\right]\right)^{\frac{1}{8}} = \left(\prod_{k=1}^{4}\mathbb{E}\left[(x^{i_j,1}_1)^8\right]\right)^{\frac{1}{8}}\left(\prod_{k=1}^{4}\mathbb{E}\left[(x^{i_j,2}_1)^8\right]\right)^{\frac{1}{8}}$$

$$\leq \sup_{i\in[r],\, j\in[2]}\mathbb{E}\left[(x^{i,j}_1)^8\right].$$

By the boundedness of $x^{i,j}$ ($i \in [r]$, $j \in [2]$), the last term is bounded uniformly in $n$, and consequently, it holds that

$$\sup_n \mathbb{E}\left[x^{i_1,1}_{\gamma_1} x^{i_2,1}_{\gamma_2} x^{i_3,1}_{\gamma_3} x^{i_4,1}_{\gamma_4} x^{i_1,2}_{\gamma_5} x^{i_2,2}_{\gamma_6} x^{i_3,2}_{\gamma_7} x^{i_4,2}_{\gamma_8}\right] < \infty.$$

Hence, we compute

$$
\begin{aligned}
\mathbb{E}(X_\alpha^4) \lesssim {} & \sum_{i_1,\ldots,i_4=1}^{r} \sum_{\gamma_1,\ldots,\gamma_8=1}^{n} \mathbb{E}\left[W^{i_1,1}_{\alpha\gamma_1} W^{i_2,1}_{\alpha\gamma_2} W^{i_3,1}_{\alpha\gamma_3} W^{i_4,1}_{\alpha\gamma_4} W^{i_1,2}_{\alpha\gamma_5} W^{i_2,2}_{\alpha\gamma_6} W^{i_3,2}_{\alpha\gamma_7} W^{i_4,2}_{\alpha\gamma_8}\right] \\
= {} & \sum_{i=1}^{r} \sum_{\gamma_1,\ldots,\gamma_8=1}^{n} \mathbb{E}\left[W^{i,1}_{\alpha\gamma_1} W^{i,1}_{\alpha\gamma_2} W^{i,1}_{\alpha\gamma_3} W^{i,1}_{\alpha\gamma_4} W^{i,2}_{\alpha\gamma_5} W^{i,2}_{\alpha\gamma_6} W^{i,2}_{\alpha\gamma_7} W^{i,2}_{\alpha\gamma_8}\right] \\
& + 4\sum_{i_1=1}^{r} \sum_{i_2\neq i_1} \sum_{\gamma_1,\ldots,\gamma_8=1}^{n} \mathbb{E}\left[W^{i_1,1}_{\alpha\gamma_1} W^{i_1,1}_{\alpha\gamma_2} W^{i_1,1}_{\alpha\gamma_3} W^{i_2,1}_{\alpha\gamma_4} W^{i_1,2}_{\alpha\gamma_5} W^{i_1,2}_{\alpha\gamma_6} W^{i_1,2}_{\alpha\gamma_7} W^{i_2,2}_{\alpha\gamma_8}\right] \\
& + 3\sum_{i_1=1}^{r} \sum_{i_2\neq i_1} \sum_{\gamma_1,\ldots,\gamma_8=1}^{n} \mathbb{E}\left[W^{i_1,1}_{\alpha\gamma_1} W^{i_1,1}_{\alpha\gamma_2} W^{i_2,1}_{\alpha\gamma_3} W^{i_2,1}_{\alpha\gamma_4} W^{i_1,2}_{\alpha\gamma_5} W^{i_1,2}_{\alpha\gamma_6} W^{i_2,2}_{\alpha\gamma_7} W^{i_2,2}_{\alpha\gamma_8}\right] \\
& + 6\sum_{i_1=1}^{r} \sum_{i_2\neq i_1} \sum_{i_3\notin\{i_1,i_2\}} \sum_{\gamma_1,\ldots,\gamma_8=1}^{n} \mathbb{E}\left[W^{i_1,1}_{\alpha\gamma_1} W^{i_1,1}_{\alpha\gamma_2} W^{i_2,1}_{\alpha\gamma_3} W^{i_3,1}_{\alpha\gamma_4} W^{i_1,2}_{\alpha\gamma_5} W^{i_1,2}_{\alpha\gamma_6} W^{i_2,2}_{\alpha\gamma_7} W^{i_3,2}_{\alpha\gamma_8}\right] \\
& + \sum_{\substack{i_1,i_2,i_3,i_4=1 \\ \text{all distinct}}}^{r} \sum_{\gamma_1,\ldots,\gamma_8=1}^{n} \mathbb{E}\left[W^{i_1,1}_{\alpha\gamma_1} W^{i_2,1}_{\alpha\gamma_2} W^{i_3,1}_{\alpha\gamma_3} W^{i_4,1}_{\alpha\gamma_4} W^{i_1,2}_{\alpha\gamma_5} W^{i_2,2}_{\alpha\gamma_6} W^{i_3,2}_{\alpha\gamma_7} W^{i_4,2}_{\alpha\gamma_8}\right] \\
=: {} & A_1 + 4A_2 + 3A_3 + 6A_4 + A_5.
\end{aligned}
$$

Define $\sigma$ by $\sigma = \max\{\sigma_{W^{i,j}} : i \in [r], \ j \in [2]\}$. Then, we compute $A_1$ as

$$
\begin{aligned}
A_1 = {} & \sum_{i=1}^{r} \sum_{\gamma_1,\ldots,\gamma_8=1}^{n} \mathbb{E}\left[W^{i,1}_{\alpha\gamma_1} W^{i,1}_{\alpha\gamma_2} W^{i,1}_{\alpha\gamma_3} W^{i,1}_{\alpha\gamma_4}\right] \mathbb{E}\left[W^{i,2}_{\alpha\gamma_5} W^{i,2}_{\alpha\gamma_6} W^{i,2}_{\alpha\gamma_7} W^{i,2}_{\alpha\gamma_8}\right] \\
= {} & \sum_{i=1}^{r} \sum_{\gamma_1,\gamma_2=1}^{n} \mathbb{E}\left[(W^{i,1}_{\alpha\gamma_1})^4\right] \mathbb{E}\left[(W^{i,2}_{\alpha\gamma_2})^2\right] \\
& + 3\sum_{i=1}^{r} \sum_{(j,j')\in J} \sum_{\gamma_1,\gamma_2=1}^{n} \sum_{\gamma_3\neq\gamma_2} \mathbb{E}\left[(W^{i,j}_{\alpha\gamma_1})^4\right] \mathbb{E}\left[(W^{i,j'}_{\alpha\gamma_2})^2\right] \mathbb{E}\left[(W^{i,j'}_{\alpha\gamma_3})^2\right] \\
& + 9\sum_{i=1}^{r} \sum_{\gamma_1,\gamma_2=1}^{n} \sum_{\gamma_3\neq\gamma_1} \sum_{\gamma_4\neq\gamma_2} \mathbb{E}\left[(W^{i,1}_{\alpha\gamma_1})^2\right] \mathbb{E}\left[(W^{i,1}_{\alpha\gamma_2})^2\right] \mathbb{E}\left[(W^{i,2}_{\alpha\gamma_3})^2\right] \mathbb{E}\left[(W^{i,2}_{\alpha\gamma_4})^2\right] \\
= {} & \sum_{i=1}^{r} \left(\frac{9\sigma^4_{W^{i,1}}\sigma^4_{W^{i,2}}}{n^2} + 3\sum_{(j,j')\in J} \frac{3(n-1)\sigma^4_{W^{i,j}}\sigma^4_{W^{i,j'}}}{n^2} + 9\frac{(n-1)^2\sigma^4_{W^{i,1}}\sigma^4_{W^{i,2}}}{n^2}\right) \\
= {} & 9\sum_{i=1}^{r} \sigma^4_{W^{i,1}}\sigma^4_{W^{i,2}} \lesssim \sigma^8.
\end{aligned}
$$

Likewise, we have

$$
\begin{aligned}
A_2 = {} & \sum_{i_1=1}^{r} \sum_{i_2\neq i_1} \sum_{(j,j')\in J} \sum_{\gamma_1,\ldots,\gamma_8=1}^{n} \mathbb{E}\left[W^{i_1,1}_{\alpha\gamma_1} W^{i_1,1}_{\alpha\gamma_2} W^{i_1,1}_{\alpha\gamma_3} W^{i_2,j}_{\alpha\gamma_4}\right] \mathbb{E}\left[W^{i_1,2}_{\alpha\gamma_5} W^{i_1,2}_{\alpha\gamma_6} W^{i_1,2}_{\alpha\gamma_7} W^{i_2,j'}_{\alpha\gamma_8}\right] d^{(i_2,j)}_{(i_1,1)} d^{(i_2,j')}_{(i_1,2)} \\
= {} & 9\sum_{i_1=1}^{r} \sum_{i_2\neq i_1} \sum_{(j,j')\in J} \sigma^4_{W^{i_1,1}}\sigma^4_{W^{i_1,2}} d^{(i_2,j)}_{(i_1,1)} d^{(i_2,j')}_{(i_1,2)} \lesssim \sigma^8
\end{aligned}
$$

and

$$A_3 = \sum_{i_1=1}^{r} \sum_{i_2 \neq i_1} \sum_{(j,j') \in J} \sum_{\gamma_1,\ldots,\gamma_8=1}^{n} \mathbb{E}\left[ W_{\alpha\gamma_1}^{i_1,1} W_{\alpha\gamma_2}^{i_1,1} W_{\alpha\gamma_3}^{i_2,j} W_{\alpha\gamma_4}^{i_2,j} \right] \mathbb{E}\left[ W_{\alpha\gamma_5}^{i_1,2} W_{\alpha\gamma_6}^{i_1,2} W_{\alpha\gamma_7}^{i_2,j'} W_{\alpha\gamma_8}^{i_2,j'} \right] d_{(i_1,1)}^{(i_2,j)} d_{(i_1,2)}^{(i_2,j')}$$

$$= 9 \sum_{i_1=1}^{r} \sum_{i_2 \neq i_1} \sum_{(j,j') \in J} \sigma_{W^{i_1,1}}^{4} \sigma_{W^{i_1,2}}^{4} d_{(i_1,1)}^{(i_2,j)} d_{(i_1,2)}^{(i_2,j')} \lesssim \sigma^8.$$

Applying a similar argument, we can also show that $A_4 \lesssim \sigma^8$ and $A_5 \lesssim \sigma^8$ holds. Therefore, we conclude that $\mathbb{E}(X_\alpha^4) \lesssim \sigma^8$ holds, which completes the proof. $\qquad\square$

**Lemma D.9.** *Take $k_1,\ldots,k_8 \in \{(i,j) : i \in [r],\ j \in [2]\}$ arbitrarily. Then, the following statements hold as $n \to \infty$.*

(i) *For $(x^{k_1},\ldots,x^{k_8})$, we have*

$$\mathbb{E}\left[\left(\frac{1}{n}\sum_{\gamma_1=1}^{n} x_{\gamma_1}^{k_1} x_{\gamma_1}^{k_2}\right)\left(\frac{1}{n}\sum_{\gamma_2=1}^{n} x_{\gamma_2}^{k_3} x_{\gamma_2}^{k_4}\right)\left(\frac{1}{n}\sum_{\gamma_3=1}^{n} x_{\gamma_3}^{k_5} x_{\gamma_3}^{k_6}\right)\left(\frac{1}{n}\sum_{\gamma_4=1}^{n} x_{\gamma_4}^{k_7} x_{\gamma_4}^{k_8}\right)\right]$$
$$\to \mathbb{E}\left[Z^{x^{k_1}} Z^{x^{k_2}}\right] \mathbb{E}\left[Z^{x^{k_3}} Z^{x^{k_4}}\right] \mathbb{E}\left[Z^{x^{k_5}} Z^{x^{k_6}}\right] \mathbb{E}\left[Z^{x^{k_7}} Z^{x^{k_8}}\right].$$

(ii) *For $(x^{k_1},\ldots,x^{k_4})$, we have*

$$\mathbb{E}\left[\left(\frac{1}{n}\sum_{\gamma_1=1}^{n} x_{\gamma_1}^{k_1} x_{\gamma_1}^{k_2}\right)\left(\frac{1}{n}\sum_{\gamma_2=1}^{n} x_{\gamma_2}^{k_3} x_{\gamma_2}^{k_4}\right)\right] \to \mathbb{E}\left[Z^{x^{k_1}} Z^{x^{k_2}}\right] \mathbb{E}\left[Z^{x^{k_3}} Z^{x^{k_4}}\right].$$

*Proof.* Define the residual term $R_\ell$ by

$$R_\ell = \frac{1}{n}\sum_{\gamma_1=1}^{n} x_{\gamma_\ell}^{k_{2\ell-1}} x_{\gamma_\ell}^{k_{2\ell}} - \mathbb{E}\left[Z^{x^{k_{2\ell-1}}} Z^{x^{k_{2\ell}}}\right]$$

for each $i \in [4]$. Define constants $C$ and $\tilde{R}$ by

$$C = \max_{\ell \in [4]} \left|\mathbb{E}\left[Z^{x^{k_{2\ell-1}}} Z^{x^{k_{2\ell}}}\right]\right|, \qquad \tilde{R} = \left(\max_{\ell \in [4]} \mathbb{E}(R_\ell^4)\right)^{\frac{1}{4}}.$$

Note that $C$ is bounded by the boundedness of $x^{i,j}$ for all $i \in [r]$ and $j \in [2]$ (see Definition 3.1). Then, we can write

$$\left|\mathbb{E}\left[\prod_{\ell=1}^{4}\left(\frac{1}{n}\sum_{\gamma_\ell=1}^{n} x_{\gamma_\ell}^{k_{2\ell-1}} x_{\gamma_\ell}^{k_{2\ell}}\right)\right] - \prod_{\ell=1}^{4}\mathbb{E}\left[Z^{x^{k_{2\ell-1}}} Z^{x^{k_{2\ell}}}\right]\right|$$
$$= \left|\mathbb{E}\left[\prod_{\ell=1}^{4}\left(R_\ell + \mathbb{E}\left[Z^{x^{k_{2\ell-1}}} Z^{x^{k_{2\ell}}}\right]\right)\right] - \prod_{\ell=1}^{4}\mathbb{E}\left[Z^{x^{k_{2\ell-1}}} Z^{x^{k_{2\ell}}}\right]\right|$$
$$\leq 4C^3\tilde{R} + 6C^2\tilde{R}^2 + 4C\tilde{R}^3 + \tilde{R}^4,$$

where the last inequality follows from the Cauchy–Schwarz inequality and the Lyapunov inequality. Likewise, we have

$$\left|\mathbb{E}\left[\prod_{\ell=1}^{2}\left(\frac{1}{n}\sum_{\gamma_\ell=1}^{n} x_{\gamma_\ell}^{k_{2\ell-1}} x_{\gamma_\ell}^{k_{2\ell}}\right)\right] - \prod_{\ell=1}^{2}\mathbb{E}\left[Z^{x^{k_{2\ell-1}}} Z^{x^{k_{2\ell}}}\right]\right|$$
$$= \left|\mathbb{E}\left[\prod_{\ell=1}^{2}\left(R_\ell + \mathbb{E}\left[Z^{x^{k_{2\ell-1}}} Z^{x^{k_{2\ell}}}\right]\right)\right] - \prod_{\ell=1}^{2}\mathbb{E}\left[Z^{x^{k_{2\ell-1}}} Z^{x^{k_{2\ell}}}\right]\right|$$
$$\leq 2C\tilde{R} + \tilde{R}^2.$$

Thus, it remains only to prove that $\tilde{R}$ converges to 0 as $n \to \infty$. This can be achieved by showing that $\mathbb{E}(R_\ell^4)$ converges to 0 for all $\ell \in [4]$. By Fact 3.1 and Lemma C.4, for all $\ell \in [4]$, we know $R_\ell$ converges almost surely to 0. The continuous mapping theorem then yields

$$R_\ell^4 \xrightarrow{a.s.} 0.$$

To upgrade this to convergence in expectation, Facts C.1 and C.2 imply it suffices to show the existence of a constant $\delta > 1$ that satisfies

$$\sup_n \mathbb{E}(R_\ell^{4+\delta}) < \infty.$$

But since each $x^{i,j}$ and its infinite-width limit $Z^{x^{i,j}}$ are bounded, such a $\delta$ exists. Therefore, we conclude that $\mathbb{E}(R_\ell^4)$ converges to 0 for all $\ell \in [4]$, and consequently, $\tilde{R}$ does as well.  $\square$

### D.1.2 $S_1$ Converges to 0

We study the convergence of the term $S_1$ defined in Section 4. Specifically, we show

$$S_1 = \left| \mathbb{E} f\left( \frac{1}{n} \sum_{\alpha=1}^n \psi(g_\alpha^{1:M}, p_{1:r}) \right) - \mathbb{E} f\left( \mathbb{E}\left[ \psi(Z^{g^{1:M}}, p_{1:r}) \mid p_{1:r} \right] \right) \right| \to 0.$$

Fix a small $\epsilon > 0$. Since $\mathring{p}_{1:r}$ is a Gaussian vector (see Definition 3.1), it is tight. Hence there exists a compact set $K \subset \mathbb{R}^r$ that satisfies

$$P(\mathring{p}_{1:r} \in K) > 1 - \epsilon.$$

Set

$$y = \frac{1}{n} \sum_{\alpha=1}^n \psi(g_\alpha^{1:M}, p_{1:r}), \quad z = \mathbb{E}\left[ \psi(Z^{g^{1:M}}, p_{1:r}) \mid p_{1:r} \right].$$

Then, we have

$$
\begin{aligned}
S_1 &= |\mathbb{E} f(y) - \mathbb{E} f(z)| \\
&\leq |\mathbb{E}[(f(y) - f(z))1\{p_{1:r} \in K\}]| + |\mathbb{E}[(f(y) - f(z))1\{p_{1:r} \notin K\}]| \\
&\leq |\mathbb{E}[(f(y) - f(z))1\{p_{1:r} \in K\}]| + \mathbb{E}[|f(y) - f(z)|1\{p_{1:r} \notin K\}].
\end{aligned}
\tag{6}
$$

By the boundedness of $f$, the second term of Eq. (6) is bounded as

$$\mathbb{E}[|f(y) - f(z)|1\{p_{1:r} \notin K\}] \leq C P(p_{1:r} \notin K)$$

with some constant $C$. Furthermore, noting that $K$ is a Borel set, applying Lemma C.2 and Proposition D.1 gives

$$\lim_{n \to \infty} P(p_{1:r} \notin K) = P(\mathring{p}_{1:r} \notin K) < \epsilon.$$

Therefore, for large enough $n$, we have

$$\mathbb{E}[|f(y) - f(z)|1\{p_{1:r} \notin K\}] < C\epsilon.$$

For the first term of Eq. (6), we have

$$
\begin{aligned}
&|\mathbb{E}[(f(y) - f(z))1\{p_{1:r} \in K\}]| \\
&\leq \sup_{a_{1:r} \in K} \left| \mathbb{E}\left[ f\left( \frac{1}{n} \sum_{\alpha=1}^n \psi(g_\alpha^{1:M}, a_{1:r}) \right) \right] - \mathbb{E}\left[ f\left( \mathbb{E}\left[ \psi(Z^{g^{1:M}}, a_{1:r}) \right] \right) \right] \right| \\
&= \sup_{a_{1:r} \in K} \Phi_n(a_{1:r}),
\end{aligned}
$$

where $\Phi_n : \mathbb{R}^r \to \mathbb{R}$ is defined by

$$\Phi_n(a_{1:r}) = \left| \mathbb{E}\left[ f\left( \frac{1}{n} \sum_{\alpha=1}^n \psi(g_\alpha^{1:M}, a_{1:r}) \right) \right] - \mathbb{E}\left[ f\left( \mathbb{E}\left[ \psi(Z^{g^{1:M}}, a_{1:r}) \right] \right) \right] \right|.$$

Define $\tilde{\psi}_{a_{1:r}} : \mathbb{R}^M \to \mathbb{R}$ by

$$\tilde{\psi}_{a_{1:r}}(u_{1:M}) = \psi(u_{1:M}, a_{1:r}). \tag{7}$$

Observe that $\tilde{\psi}_{a_{1:r}}$ is bounded by the boundedness of $\psi$. We later show that $\tilde{\psi}_{a_{1:r}}$ is also pseudo-Lipschitz (Lemma D.10). Therefore, by Eq. (3) and Lemma C.2, we have

$$\lim_{n\to\infty} \Phi_n(a_{1:r}) = 0 \quad (a_{1:r} \in \mathbb{R}^r).$$

Moreover, noting that $f$ and $\psi$ are bounded and continuous (see Fact C.3), we can apply the (uncoditional and conditional) bounded convergence theorem to show that $\Phi_n$ is continuous at every point. Therefore, we can apply Lemma D.11 to show that $\sup_{a_{1:r}\in K} \Phi_n(a_{1:r})$ converges to 0 as $n \to \infty$.

**Lemma D.10.** *Define* $\tilde{\psi}_{a_{1:r}} : \mathbb{R}^M \to \mathbb{R}$ *by Eq. (7). Then, it is pseudo-Lipschitz.*

*Proof.* Suppose $\psi$ is pseudo-Lipschitz of order $d + 1$ with $d \geq 1$. Then, we have

$$|\tilde{\psi}_{a_{1:r}}(u_{1:M}) - \tilde{\psi}_{a_{1:r}}(u'_{1:M})| = |\psi(u_{1:M}, a_{1:r}) - \psi(u'_{1:M}, a_{1:r})|$$
$$\lesssim \|(u_{1:M}) - (u'_{1:M})\|(1 + \|(u_{1:M}, a_{1:r})\|^d + \|(u'_{1:M}, a_{1:r})\|^d).$$

By Lemma C.1, we can bound $\|(u_{1:M}, a_{1:r})\|^d$ as

$$\|(u_{1:M}, a_{1:r})\|^d \leq (\|(u_{1:M})\| + \|a_{1:r}\|)^d \leq 2^{d-1}\left(\|(u_{1:M})\|^d + \|a_{1:r}\|^d\right).$$

Thus, we have

$$1 + \|(u_{1:M}, a_{1:r})\|^d + \|(u'_{1:M}, a_{1:r})\|^d \leq 1 + 2^{d-1}\left(\|(u_{1:M})\|^d + \|(u'_{1:M})\|^d\right) + 2^d\|a_{1:r}\|^d$$
$$\lesssim 1 + \|(u_{1:M})\|^d + \|(u'_{1:M})\|^d,$$

where the last inequality holds because $a_{1:r}$ is fixed. $\square$

**Lemma D.11.** *Let $K \subset \mathbb{R}^r$ be a compact set. Suppose for each $n \in \mathbb{N}$, $f_n : \mathbb{R} \to \mathbb{R}$ is a continuous function that satisfies*

$$\lim_{n\to\infty} f_n(a_{1:r}) = 0 \tag{8}$$

*for any constant $a_{1:r} \in K$. Then, we have*

$$\lim_{n\to\infty} \sup_{a_{1:r}\in K} |f_n(a_{1:r})| = 0.$$

*Proof.* Fix $\epsilon > 0$. Let $B(a_{1:r}, \epsilon)$ denote the ball of radius $\epsilon$ centered at $a_{1:r} \in \mathbb{R}^r$. By the continuity of $f_n$, for each $a_{1:r} \in K$, there exists $\delta_{a_{1:r}} > 0$ such that

$$|f_n(a_{1:r}) - f_n(b_{1:r})| < \epsilon/2 \quad (b_{1:r} \in B(a_{1:r}, \delta_{a_{1:r}}))$$

holds. Note that $K$ is covered by the union of balls as

$$K \subset \bigcup_{a_{1:r}\in K} B(a_{1:r}, \delta_{a_{1:r}}).$$

Since $K$ is compact, we can cover $K$ with finitely many balls, say

$$K \subset \bigcup_{i=1}^{I} B_i = \bigcup_{i=1}^{I} B(a_{1:r}^i, \delta_{a_{1:r}^i}),$$

where $B_i$ is given by $B_i = B(a_{1:r}^i, \delta_{a^i})$. By Eq. (8), for each $i \in [I]$, there exists $N_i \in \mathbb{N}$ such that

$$|f_n(a_{1:r}^i)| < \epsilon/2 \quad (n \geq N_i)$$

holds. Let $N$ denote the maximum of $\{N_i : i \in [I]\}$. Then, for any $n \geq N$, we have

$$\sup_{a_{1:r}\in K} |f_n(a_{1:r})| \leq \max_{i\in[I]} \sup_{a_{1:r}\in B_i} |f_n(a_{1:r})| \leq \max_{i\in[I]} \sup_{a_{1:r}\in B_i} (|f_n(a_{1:r}) - f_n(a_{1:r}^i)| + |f_n(a_{1:r}^i)|)$$
$$< \max_{i\in[I]} \epsilon = \epsilon,$$

which implies the convergence of $\sup_{a_{1:r}\in K} |f_n(a_{1:r})|$ to zero as $n$ goes to infinity. $\square$

### D.1.3 $S_2$ Converges to 0

We study the convergence of the term $S_2$ also defined in Section 4. Specifically, we prove

$$S_2 = \left| \mathbb{E}f\left(\mathbb{E}\left[\psi(Z^{g^{1:M}}, \boldsymbol{p}_{1:r}) \mid \boldsymbol{p}_{1:r}\right]\right) - \mathbb{E}f\left(\mathbb{E}[\psi(Z^{g^{1:M}}, \mathring{\boldsymbol{p}}_{1:r}) \mid \mathring{\boldsymbol{p}}_{1:r}]\right) \right| \to 0.$$

Define a function $\Psi : \mathbb{R}^r \to \mathbb{R}$ by

$$\Psi(\boldsymbol{a}_{1:r}) = \mathbb{E}\left[\psi(Z^{g^{1:M}}, \boldsymbol{a}_{1:r})\right],$$

then it is bounded since $\psi$ is bounded. By the bounded convergence theorem, it is also continuous. Therefore, by the continuous mapping theorem, we have

$$\Psi(\boldsymbol{p}_{1:r}) \xrightarrow{d} \Psi(\mathring{\boldsymbol{p}}_{1:r}).$$

Since $Z^{g^{1:M}}$ is independent of $\boldsymbol{p}_{1:r}$, we have

$$\mathbb{E}\left[\psi(Z^{g^{1:M}}, \boldsymbol{p}_{1:r}) \mid \boldsymbol{p}_{1:r}\right] = \Psi(\boldsymbol{p}_{1:r}).$$

Therefore, we have

$$\mathbb{E}\left[\psi(Z^{g^{1:M}}, \boldsymbol{p}_{1:r}) \mid \boldsymbol{p}_{1:r}\right] \xrightarrow{d} \Psi(\mathring{\boldsymbol{p}}_{1:r}).$$

Note that $\Psi(\mathring{\boldsymbol{p}}_{1:r})$ can be expressed as

$$\Psi(\mathring{\boldsymbol{p}}_{1:r}) = \mathbb{E}\left[\psi(Z^{g^{1:M}}, \mathring{\boldsymbol{p}}_{1:r}) \mid \mathring{\boldsymbol{p}}_{1:r}\right],$$

where $Z^{g^{1:M}}$ is independent of $\mathring{\boldsymbol{p}}_{1:r}$. By Lemma C.2, this implies $S_2 \to 0$.

### D.2 Proof of Corollary 3.2

Let $\psi : \mathbb{R}^J \to \mathbb{R}$ be a bounded and Lipschitz function that satisfies $|\psi| \le C$. Note that by Fact C.3, it is also pseudo-Lipschitz. Define a bounded and continuous function $f : \mathbb{R} \to \mathbb{R}$ by

$$f(x) = -C\mathbf{1}\{x < C\} + x\mathbf{1}\{x \in [-C, C]\} + C\{x \ge C\}.$$

Then, Lemma C.2 and Theorem 3.1 imply that

$$\mathbb{E}[\psi(h_\alpha^1, \ldots, h_\alpha^J)] = \mathbb{E}\left(\frac{1}{n}\sum_{\alpha=1}^n \psi(h_\alpha^1, \ldots, h_\alpha^J)\right) = \mathbb{E}\left[f\left(\frac{1}{n}\sum_{\alpha=1}^n \psi(h_\alpha^1, \ldots, h_\alpha^J)\right)\right]$$

$$\to \mathbb{E}\left[f\left(\mathbb{E}[\psi(Z^{h^1}, \ldots, Z^{h^J}) \mid \mathring{p}_1, \ldots, \mathring{p}_r]\right)\right] = \mathbb{E}\left[\psi(Z^{h^1}, \ldots, Z^{h^J})\right]$$

holds as $n \to \infty$. Since the above convergence holds for all bounded and Lipschitz function $\psi$, by Lemma C.2, this implies the desired convergence in distribution.

## E Sub-Gaussianity of the Attention Outputs

In this section, we discuss the sub-Gaussianity of the limiting distribution of the attention outputs from Example 3.1. In this appendix, we provide a proof of the sub-Gaussianity of the single random variable $Z^{y^i}$ for any $i \in [s]$, and consider the single-head attention setting (i.e. $H = 1$) for simplicity. Specifically, we consider

$$Z^{y^i} = \sum_{j=1}^s \mathrm{SoftMax}_j(\mathring{p}_{i,1}^{(1)}, \ldots, \mathring{p}_{i,s}^{(1)})Z^{\tilde{v}^{1,j}}.$$

The same proof strategy also applies to prove the sub-Gaussianity of the vector $(Z^{y^1}, \ldots, Z^{y^s})$ with $H$ being any positive integer.

For each $j \in [s]$, define $a_j(\mathring{p}_{i,1:s}^{(1)})$ by

$$a_j(\mathring{p}_{i,1:s}^{(1)}) = \mathrm{SoftMax}_j(\mathring{p}_{i,1}^{(1)}, \ldots, \mathring{p}_{i,s}^{(1)}).$$

It follows that
$$Z^{y^i} | \mathring{\boldsymbol{p}}_{i,1:s}^{(1)} \sim \mathcal{N}(0, \boldsymbol{a}_{1:s}(\mathring{\boldsymbol{p}}_{i,1:s}^{(1)})^\top \mathrm{Var}(\boldsymbol{Z}^{\tilde{v}^{1,1:s}}) \boldsymbol{a}_{1:s}(\mathring{\boldsymbol{p}}_{i,1:s}^{(1)})).$$

Since each $a_j(\mathring{\boldsymbol{p}}_{i,1:s}^{(1)})$ lies in the interval $[0, 1]$, we have

$$\|\boldsymbol{a}_{1:s}(\mathring{\boldsymbol{p}}_{i,1:s}^{(1)})\|^2 \le \sum_{j=1}^{s} a_j(\mathring{\boldsymbol{p}}_{i,1:s}^{(1)}) = 1$$

and therefore

$$\boldsymbol{a}_{1:s}(\mathring{\boldsymbol{p}}_{i,1:s}^{(1)})^\top \mathrm{Var}(\boldsymbol{Z}^{\tilde{v}^{1,1:s}}) \boldsymbol{a}_{1:s}(\mathring{\boldsymbol{p}}_{i,1:s}^{(1)}) \le \|\boldsymbol{a}_{1:s}(\mathring{\boldsymbol{p}}_{i,1:s}^{(1)})\|^2 \|\mathrm{Var}(\boldsymbol{Z}^{\tilde{v}^{1,1:s}})\|_2 \le \|\mathrm{Var}(\boldsymbol{Z}^{\tilde{v}^{1,1:s}})\|_2,$$

where $\| \cdot \|_2$ is the operator norm. As a result, for any $t \ge 0$, it holds that

$$\begin{aligned}
\mathrm{P}(|Z^{y^i}| \ge t) &= \mathbb{E}\left[\mathrm{P}(|Z^{y^i}| \ge t \mid \mathring{\boldsymbol{p}}_{i,1:s}^{(1)})\right] \\
&\le 2\mathbb{E}\left[\exp\left(-\frac{t^2}{2\boldsymbol{a}_{1:s}(\mathring{\boldsymbol{p}}_{i,1:s}^{(1)})^\top \mathrm{Var}(\boldsymbol{Z}^{\tilde{v}^{1,1:s}})\boldsymbol{a}_{1:s}(\mathring{\boldsymbol{p}}_{i,1:s}^{(1)})}\right) \Big| \mathring{\boldsymbol{p}}_{i,1:s}^{(1)}\right] \\
&\le 2\exp\left(-\frac{t^2}{2\|\mathrm{Var}(\boldsymbol{Z}^{\tilde{v}^{1,1:s}})\|_2}\right)
\end{aligned}$$

by the conditional sub-Gaussianity of $Z^{y^i}$ given $\mathring{\boldsymbol{p}}_{i,1:s}^{(1)}$.

In this argument, we can also show the sub-Gaussianity of the limiting distribution of attention outputs of the form $y^i = \frac{1}{\sqrt{H}} \sum_{a=1}^{H} \sum_{j=1}^{s} \phi_j(p_{i,1}^{(a)}, \dots, p_{i,s}^{(a)}) \tilde{v}^{a,j}$, where the softmax functions are replaced by bounded and pseudo-Lipschitz nonlinearities $\phi_j$.

