# OpenReview forum: "Infinite-Width Limit of a Single Attention Layer: Analysis via Tensor Programs"
_NeurIPS.cc/2025/Conference — NeurIPS 2025 poster_

### Official Review · Reviewer_9XUB · 2025-06-30

**Clarity:** 3
**Significance:** 2
**Originality:** 3
**Rating:** 5
**Confidence:** 3

**Summary:**

This paper studies the distribution of attention layer outputs at random initialization. Unlike previous settings (such as standard MLPs or CNNs) where the limiting distribution of preactivations becomes (at infinite width) Gaussian with variance controlled by a moment of the previous layer's distribution, the authors prove that the distribution for an attention layer at large $n$ is only Gaussian conditional on the dot product scores $p = \frac{1}{\sqrt{n}} k \cdot q$ where $k$ is key and $q$ is query. As a consequence the full distribution of the attention outputs can be non-Gaussian but with a simple hierarchical distribution that the authors characterize. They prove that the $p$ variables converge to Gaussian variables and

**Questions:**

1. Do the authors have thoughts on training dynamics at large $n$? The $\mu$P scaling of Yang et al suggests that $1/n$ attention layer scaling would admit stable training if you desire constant scale changes to keys and queries. While this doesn't have an interesting large $n$ limit at initialization, it is becoming more commonly employed in practical LLM training.
2. I recommend the authors to define
$$y = \frac{1}{\sqrt H} \sum_{a=1}^H  v^{ a } \text{Softmax}( p^{a} )$$
, since when you include the $\frac{1}{\sqrt H}$ it keeps the variance stable as heads $H$ is increased and would make Figure 2b's comparison more reasonable.
3. The authors claim that heads are not infinite and that this makes large head limits unrealistic. To be fair, the $n$ is not infinite in practice either. Wouldn't it be more reasonable to study which dimensions are **scaled up** as practitioners vary model size? In many papers (such as the LLAMA paper  https://arxiv.org/abs/2302.13971 Table 2 , or the GPT paper https://arxiv.org/abs/2005.14165 Table 2.1), the $n$ (dimension per head) is either not scaled up or is scaled up much more slowly than the number of heads.

This work misses some key papers that have studied large $n$ limits of transformers, both at initialization and during training
[1] https://arxiv.org/abs/2403.02579 which study the signal propagation of deep transformers with the $1/\sqrt{n}$ scaling at initialization.

[2] https://arxiv.org/abs/2405.15712, which studies the training dynamics of infinite $n$, infinite $H$, and infinite depth $L$ limits of transformers under different scalings. They argue that the large $n$ limit with $1/\sqrt{n}$ has an interesting initial distribution, it does not admit stable training. If you adopt a $\frac{1}{n}$ scaling, then it allows stable training but taking $n \to \infty$ causes all heads to collapse to the same values. Instead the large head limit at fixed $N$ maintains diversity across attention heads.

[3] https://arxiv.org/abs/2306.17759, where the authors study the signal propagation of *deep shaped transformers* where the depth and width diverge proportionally and the softmax attention is rescaled. In this paper the authors derive a stochastic differential equation for the evolution of the cross-token correlation along the residual stream of the network.

It would be helpful to the reader to understand the similarities / differences between their approach and the findings of these other works.

In general, I am very open to increasing my score upon some of these issues being resolved.

**Ethical Concerns:**

["NO or VERY MINOR ethics concerns only"]

**Final Justification:**

The authors analyze an important problem (the feature distribution of randomly initialized transformers at various head counts and head dimensions). With their promised revisions and clarifications of the limitations of their approach, I am in support of acceptance of this work.

**Limitations:**

The authors could expand do acknowledge that this work focuses on the initialization regime. They could consider adding information that they currently require samples & sequence length to be fixed as $n \to \infty$.

**Quality:**

3

**Strengths And Weaknesses:**

Strengths:
1. The result is very intuitive and the authors provide rigorous derivations which establish this limiting distribution.
2. The authors also provide a few experiments that show the validity of their results.

Weaknesses:
1. The authors focus on properties of the transformer at random initialization. The authors do not argue or show that this scaling rule yields stable training in the $n\to\infty$ limit (indeed prior works showed that the $1/n$ scaling is necessary if $k, q$ variables move by $\Theta(1)$ since their updates are correlated https://arxiv.org/abs/2203.03466, https://arxiv.org/abs/2405.15712).
2. The analysis also requires batchsize and sequence length (total tokens) to be constant with respect to $n$ as $n \to \infty$. While this is commonly employed in signal propagation papers, it could be disregarding important effects when describing modern transformers.
3. One weakness is that the paper may be significance / novelty in that large $n$ and large $H$ limit were analyzed in Hron et al. This paper removes the need for a large $H$ limit, but at the cost of a more complicated hierarchical distribution for the output variables. However, I am willing to improve my assessment of novelty / significance during conversations with the authors.

---

> ### Author Rebuttal · Authors · 2025-07-31
>
> Thank you for the thoughtful review and for highlighting important limitations and additional literature.
>
> **Weakness 1 and Q1: Training dynamics.**
>
> **Reply:** Thank you for your insightful question. We agree that understanding the conditions for stable training is a vital research direction.
>
> Our choice of $1/\sqrt{n}$-scaling is deliberate, as it reflects the standard used not only in the vast majority of practical Transformer implementations but also in much of the relevant theoretical literature (e.g., Hron et al., 2020; Dinan et al., 2023; Cowsik et al., 2024). We believe that establishing a rigorous theoretical foundation for this canonical setting is an indispensable first step.
>
> Furthermore, the definitive theory for what ensures stable training is still an evolving area of research (like Bordelon et al., 2024), and may depend on factors not typically considered in standard infinite-width limits. For instance, recent work has shown that data statistics can significantly alter the optimal scaling rules of standard muP (Hayou and Liu, 2025).
>
> The theoretical understanding of self-attention is a collective effort, with different limits (e.g., $1/n$, $1/\sqrt{n}$, infinite-head) each providing valuable insights. Our contribution is to provide the first rigorous characterization of the initial distribution for the most commonly used scaling. We see this as providing the essential, previously missing groundwork for future research on the training stability of these widely-used models.
>
> **Weakness 2: Fixed batch size and sequence length.**
>
> **Reply:** Thank you for this important observation. As the reviewer correctly notes, our analysis assumes a constant sequence length and batch size as $n\to\infty$. This is a standard assumption made in the existing Tensor Programs literature (Yang, 2019), which our work builds upon, to keep the theoretical analysis tractable. We acknowledge that modern Transformers often utilize very long sequences, and our setting may not capture all the dynamics present in such cases. As suggested, we will expand our limitations section in the revised manuscript to more explicitly state this assumption and its potential implications. Extending the framework to handle growing sequence lengths is a challenging but important direction for future research.
>
> **Weakness 3: Novelty and significance.**
>
> **Reply:** Thank you for this comment and for the opportunity to clarify the novelty and significance of our work in relation to Hron et al. (2020).
>
> As the reviewer notes, Hron et al. (2020) analyzed the large-$n$ and large-$H$ limit. Importantly, they also empirically observed that the finite-head case exhibits non-Gaussian behavior, which poses a major challenge for the standard NNGP framework. This observation motivated their shift to the analytically more tractable infinite-head limit, effectively sidestepping the question of what the exact distribution looks like in realistic finite-head settings.
>
> Our contribution directly addresses this open problem by providing the first rigorous derivation of the limiting distribution in the finite-head regime under standard $1/\sqrt{n}$-scaling. We show that the limiting distribution is a hierarchical Gaussian, which, although more complex than a standard Gaussian, is an exact characterization and captures important aspects of real-world attention mechanisms that previous Gaussian approximations cannot.
>
> Furthermore, we believe this result opens the door to a deeper theoretical understanding of Transformers. In particular, the hierarchical Gaussian structure we uncover at the first layer suggests a pathway to study how non-Gaussianity may compound through depth, possibly leading to heavy-tailed or anisotropic distributions in deep layers. These phenomena could play a critical role in the representational power and feature learning dynamics of Transformers. We see our work as a first rigorous step toward such a refined theory, one that moves beyond infinite-width approximations and closer to practical architectures.
>
> Finally, we would be very happy to engage in further discussion if there are particular directions or questions the reviewer is especially interested in. We welcome any follow-up comments or suggestions.
>
> **Q2: The attention scaling considering head counts.**
>
> **Reply:** Thank you for this excellent and highly constructive suggestion. We agree completely that including the $1/\sqrt{H}$-scaling factor provides a more meaningful comparison as the number of heads increases. We will adopt this formulation throughout our revised manuscript and have redone the corresponding experiments.
>
> This change has significantly strengthened our work by enabling a direct comparison with Hron et al. (2020). We found that with this scaling:
>
> - The limiting distribution of the attention scores in our framework perfectly matches the Gaussian distribution in their infinite-head limit.
> - The variance of our finite-head non-Gaussian output distribution theoretically matches the variance of their infinite-head Gaussian output distribution.
>
> To illustrate this, we reran the experiment for Figure 2(b) and included the theoretical distribution from Hron et al. (2020).  The new plot, which we will include in the revised manuscript, now visually demonstrates that as $H$ increases, our non-Gaussian distribution approaches the Gaussian distribution from Hron et al. (2020). This observation provides strong visual support for our framework as a more general theory that encompasses the infinite-head limit.
>
> **Q3: The realism of the theoretical setting.**
>
> **Reply:** This is a very important point. We agree that many state-of-the-art models scale up the number of heads $H$ significantly. Our response is twofold.
>
> First, the main insight of our work is a characterization of the attention mechanism that is broadly applicable, regardless of the number of heads $H$. While our theory is formally derived in the $n\to\infty$ limit, its predictions are not confined to a specific relative scaling speed of $n$ versus $H$. Our experiments confirm this robustness across the practical scenarios. Figure 2(b) shows that our theory remains accurate even as $H$ grows large, and the analysis in our appendix (Figure 3) further validates our framework in the common setting where the dimension-per-head is fixed.
>
> Second, we believe that a rigorous analysis of the finite-$H$ regime remains valuable. It provides a foundational understanding of the attention mechanism's building blocks and is directly applicable to smaller-scale models where the infinite-head approximation may not be suitable.
>
> We will revise our manuscript to emphasize more clearly that our findings provide a robust description for any number of heads, clarifying their broad applicability.
>
> **On missed citations.**
>
> **Reply:** We thank you for bringing these key papers to our attention. We will cite and discuss these works in our revised manuscript. The primary differences between their approaches and ours lie in the scope and methodology:
>
> - Cowsik et al. (2024) use mean-field theory to characterize the edge of chaos by analyzing forward/backward signal propagation. Their analysis, however, relies on several simplifying assumptions (e.g., the Gaussianity of the QK product) to make the problem tractable. Our work, in contrast, provides a rigorous derivation of the exact limiting distribution of the forward pass without such assumptions.
> - Bordelon et al. (2024) use dynamical mean field theory to provide an important analysis of the training dynamics of Transformers under various infinite limits, including infinite width, heads, and depth. Their primary focus is on identifying parameterizations that ensure stable feature learning over time. While their analysis focuses on the time evolution of second moments (kernels), it does not provide a rigorous derivation of the exact limiting output distribution at initialization. Our work, with its distinct scope, addresses this specific problem by establishing the precise, non-Gaussian initial distribution.
> - Noci et al. (2023) aim to describe signal evolution using a stochastic differential equation, which requires modifying the standard softmax function. Our scope is distinct, as we focus on rigorously analyzing the attention layer in its standard, unmodified form.
>
> **Reference**
>
> [1] Bordelon et al. (2024). Infinite Limits of Multi-head Transformer Dynamics.
>
> [2] Cowsik et al. (2024). Geometric dynamics of signal propagation predict trainability of transformers.
>
> [3] Dinan et al. (2023). Effective Theory of Transformers at Initialization.
>
> [4] Hayou and Liu (2025). Optimal Embedding Learning Rate in LLMs: The Effect of Vocabulary Size.
>
> [5] Hron et al. (2020). Infinite attention: NNGP and NTK for deep attention networks.
>
> [6] Noci et al. (2023). The Shaped Transformer: Attention Models in the Infinite Depth-and-Width Limit.
>
> [7] Yang (2019). Scaling limits of wide neural networks with weight sharing: Gaussian process behavior, gradient independence, and neural tangent kernel derivation.

---

> > ### Comment · Reviewer_9XUB · 2025-08-04
> >
> > I thank the authors for their detailed rebuttal and their attempts to improve the paper by fixing Fig 2b, improving the completeness of their references and comparisons to prior work, and expanding their limitations section. Overall I think this work is a useful step towards understanding the internal feature distributions of randomly initialized transformers and will raise my score to reflect this.

---

### Official Review · Reviewer_NP2J · 2025-06-30

**Clarity:** 2
**Significance:** 3
**Originality:** 2
**Rating:** 4
**Confidence:** 3

**Summary:**

The paper studies the infinite-width limit of a single attention layer by extending the Tensor Programs framework to handle the standard $1 / \sqrt{n}$ attention scaling used in transformers. The authors find that in comparison to previously studied limits which use a $1 / n$ scaling, the output distribution is non-Gaussian due to a hierarchical structure where the output is Gaussian when conditioned on the jointly Gaussian random attention scores. The authors then provide some numerical experiments demonstrating the accuracy of the limiting distribution for finite widths.

**Questions:**

I am a bit confused about the proposed scaling regime and its relations to other works.

1. muP scaling [1]: My understanding is that the $1/n$ scaling (as opposed to $1/\sqrt{n}$ scaling) is *necessary* to prevent blow-up after one step. I believe that [2] also supports this claim. Will this be an issue in the scaling of this paper?

2. Both [2] and [3] also analyze the infinite head limit. But [2] seems to make the head dimension finite, while [3] (like in this paper) also grows the head dimension. Is there a fundamental difference here to be aware of? It also seems that in practice the number of heads is very large, so is the infinite-head limit actually a bad approximation?

3. What is the relation of all these works to the scaling in [4]? It seems there is also a $1/ \sqrt{n}$ scaling studied here?

As a comment, I think [3], [4] should be cited and discussed.

1. Yang et al. (2021), Tuning large neural networks via zero-shot hyperparameter transfer.
2. Bordelon et al. (2024), Infinite Limits of Multi-head Transformer Dynamics
3. Hron et al. (2020), Infinite attention: NNGP and NTK for deep attention networks
4. Dinan et al. (2023), Effective Theory of Transformers at Initialization.

**Ethical Concerns:**

["NO or VERY MINOR ethics concerns only"]

**Final Justification:**

I believe the setting is interesting due to the foundation nature and the results are solid. However, I am still not completely convinced that this setting is very important due to the fact that it is limited to one layer, has theoretical instabilities after training, may be well approximated by the infinite head limit.

**Limitations:**

Yes.

**Quality:**

3

**Strengths And Weaknesses:**

The results is novel and interesting because it characterizes a different infinite-width limit for a practically relevant instantiation of attention. However, the result is limited to only a single attention layer and only describes the forward pass at initialization.

---

> ### Author Rebuttal · Authors · 2025-07-31
>
> Thank you for the constructive feedback and for suggesting additional literature.
>
> **Weakness: On the limitation to a single layer at initialization.**
>
> **Reply:** Thank you for the comment. This is indeed a valid and important point.
> We intentionally focused on a single attention layer and forward pass at initialization, as even this setting has remained analytically intractable under realistic conditions such as finite heads and standard $1/\sqrt{n}$-scaling. Our goal was to rigorously characterize the behavior of attention without relying on approximations like infinite-head limits or altered scaling, which are common in prior theoretical work.
>
> We view this contribution as a foundational step, analogous to how the development of the Neural Network Gaussian Process (NNGP) framework first established forward-pass analyses for fully connected networks before later extensions to training dynamics and deeper architectures. Similarly, a precise and rigorous understanding of a single layer is essential before tackling the more complex and layered nature of modern Transformers, especially since attention layers exhibit non-Gaussian behavior even at initialization.
>
> **Q1: MuP scaling.**
>
> **Reply:** Thank you for this important question regarding training stability. Our work analyzes the $1/\sqrt{n}$-scaling, a choice motivated by the fact that it is the standard used in the vast majority of practical Transformer implementations. This scaling is also a central object of study in much of the relevant theoretical literature  (e.g., Hron et al., 2020; Dinan et al., 2023; Cowsik et al., 2024).
>
> While some analyses suggest potential instabilities with this scaling, the precise conditions for stable training are still an active area of research and may depend on factors often abstracted away in simplified theoretical limits. These include the number of tokens, the use of finite low-rank QK projections, and data-specific statistics. Indeed, recent work of Hayou and Liu (2025)  has shown that data statistics can significantly alter optimal scaling rules of standard muP.
>
> The theoretical foundation of self-attention is still under active development, and we believe different limiting procedures (e.g., $1/n$, $1/\sqrt{n}$, infinite-head) each offer valuable insights into different phenomena. Our contribution to this ongoing effort is to provide the first rigorous characterization of the initial distribution for the most commonly used scaling. This establishes an essential and previously missing foundational starting point from which to analyze the more complex training dynamics.
>
> **Q2: Infinite-head limit.**
>
> **Reply:** Thank you for this insightful question. The fundamental difference is that while the infinite-head limit yields a Gaussian approximation, our work rigorously shows that for any finite number of heads, the limiting distribution is fundamentally non-Gaussian.
>
> To your second point, we agree that for many modern models, the infinite-head limit can be a reasonable approximation. Our claim is not that this limit is always bad, but that our framework offers a more general and precise perspective. In other words, while the infinite-head approximation is only accurate when the number of heads is large, our theory provides a good characterization regardless of whether the number of heads is infinite or finite. As our experiments show, our theory remains highly accurate even as $H$ grows large (Fig. 2(b)) and when $n$ and $H$ grow proportionally (Fig. 3 in the appendix). This demonstrates our theory's ability to accurately capture the behavior of attention mechanisms regardless of the number of heads.
>
> Based on your comments we will emphasize this important suggestion more in the main text.
>
> **Q3: Relation with Dinan et al. (2023).**
>
> **Reply:** Thank you for highlighting this important and relevant work. You are correct that Dinan et al. also analyze Transformers under a similar $1/\sqrt{n}$-scaling. The primary difference lies in the scope and granularity of the analysis:
>
> - Dinan et al. (2023) focus on tracking the first two moments (the kernel) of the signal to characterize its propagation properties at initialization and during training.
> - Our work, in contrast, rigorously derives the full limiting distribution of the attention layer's output, going beyond just the first two moments.
>
> This distinction is crucial because our approach reveals the fundamentally non-Gaussian nature of the output, a detail that an analysis focused only on the kernel does not capture.
>
> **On missed citations.**
>
> **Reply:** Thank you for your suggestions on citations. We will ensure our discussion of the related literature is comprehensive in the revised manuscript.
>
> - Regarding Hron et al. (2020), we would like to clarify that this key work was already cited and discussed in our paper, particularly in the context of the infinite-head Gaussian approximation.
> - Regarding Dinan et al. (2023), we agree it is highly relevant. As detailed in our response to your Question 3, our work differs in that we rigorously derive the full limiting distribution of the attention output, whereas their analysis focuses on the kernel.
> - We also agree that the work of Bordelon et al. (2024) is an important point of comparison. Their work analyzes the training dynamics under various infinite limits to find stable parameterizations. Our scope is distinct, as we focus on rigorously deriving the exact distribution at initialization, providing a foundational starting point for such dynamical analyses.
>
> **Reference**
>
> [1] Bordelon et al. (2024). Infinite Limits of Multi-head Transformer Dynamics.
>
> [2] Cowsik et al. (2024). Geometric dynamics of signal propagation predict trainability of transformers.
>
> [3] Dinan et al. (2023). Effective Theory of Transformers at Initialization.
>
> [4] Hayou and Liu (2025). Optimal Embedding Learning Rate in LLMs: The Effect of Vocabulary Size.
>
> [5] Hron et al. (2020). Infinite attention: NNGP and NTK for deep attention networks.

---

> > ### Comment · Reviewer_NP2J · 2025-08-06
> >
> > Thank you for providing these clarifications. I plan to maintain my score and increase the confidence.

---

### Official Review · Reviewer_Tp28 · 2025-07-02

**Clarity:** 2
**Significance:** 3
**Originality:** 3
**Rating:** 4
**Confidence:** 3

**Summary:**

The paper presents an analysis of the infinite width limit for transformers with $1/\sqrt{n}$ scaling of random weights, improving prior analyses that had to resort to the less interesting $1/n$ scaling. Moreover, the  paper aims to avoid a limit where the number of attention heads is taken to infinity. The arguments are viable in my view as otherwise the limit case diverges from what is observed in large - but implementable - networks.

The paper makes use of (and extends) the existing tensor program framework, which is well established in the literature. Compared to previous work, this yields meaningful results for the case of finite width and a finite number of heads. The key result is theorem 3.1, which handles the limiting distributions of pre-activations (Gaussians) as per previous results, but extends these by deriving limiting distributions for the (pre-softmax) attention inner products.  As far as I understand the key idea is to calculate these directly instead of going through the limiting distribution of the Gaussian pre-activations. This is - in principle - a viable extension of the so-called Nestor master theorem. A sketch of the theorem is given, but the proof is rather technical. An example for the multi-head case is added to illustrate the use of the novel analysis.

**Questions:**

Can you provide stronger experimental evidence to support your analysis?
What is the impact of your analysis for optimization, initialization or hyper-parameter selection?
Is there a way you can make the notation more readable? Currently it requires quite some effort.

**Ethical Concerns:**

["NO or VERY MINOR ethics concerns only"]

**Limitations:**

Yes.

**Paper Formatting Concerns:**

None.

**Quality:**

3

**Strengths And Weaknesses:**

+ Extends our understanding of infinite-limit analysis to attention-based models
+ Innovative way of handling inner products of random variables (attention pre-activation)
+ Focuses on key limitations of existing analyses
+ Provides a non-trivial technical result
- Derivation is somewhat obscure and difficult to follow (at least for non-specialists)
- Impact of analysis is not clear, what can be learned?
- Experiments are very rudimentary

---

> ### Author Rebuttal · Authors · 2025-07-31
>
> Thank you for your detailed review. We acknowledge the concerns regarding clarity, impact, and the scope of our experiments.
>
> **Weaknesses: Derivation is somewhat obscure and difficult to follow (at least for non-specialists).**
>
> **Reply:** Thank you for your valuable feedback. As suggested by Reviewer pKBC, we will add a pseudocode example to the appendix to provide a more concrete and intuitive illustration of our setup.
>
> **Weaknesses: Impact of analysis is not clear, what can be learned?**
>
> **Question: What is the impact of your analysis for optimization, initialization or hyper-parameter selection?**
>
> **Reply:** Thank you for this important question. We present specific impacts on current choices as well as future impacts.
>
> As our current work focuses on the forward pass analysis, one concrete impact is that it sheds light on how initialization scale affects the distribution of attention outputs, particularly in realistic settings with finite heads and $1/\sqrt{n}$-scaling. Understanding the output distribution precisely (e.g., its variance and tail behavior) provides a theoretical basis for choosing initialization scales that preserve signal variability and avoid pathological behaviors such as rank collapse. While prior work has offered valuable insights, it often relies on specific regimes, such as infinite heads or tailored scaling, which depart from standard practices. Our contribution is to rigorously derive the infinite-width limit under the commonly used $1/\sqrt{n}$-scaling with finite heads, revealing its non-Gaussian nature.
>
> Looking forward, we believe our framework can serve as a foundation for analyzing backward pass dynamics as well. Extending the analysis to include gradients and weight updates could eventually inform learning rate selection or adaptive scaling strategies in training attention-based models. Much like how Yang (2019) provided the foundation for later studies on training dynamics and hyperparameter selection (Yang and Hu, 2021) for neural networks without attention, we believe our work provides a similar foundation for the attention case. This remains an exciting direction for future work.
>
> We agree that the contributions of our analysis should be made more explicit, and we will revise the paper accordingly to emphasize them.
>
> **Weaknesses: Experiments are very rudimentary.**
>
> **Question: Can you provide stronger experimental evidence to support your analysis?**
>
> **Reply:** Thank you for the suggestion to include more experiments. In response, we have conducted an additional experiment by varying the spatial dimension, $s$ (the number of tokens).
>
> The table below presents the results of the experiment from Section 5.1 with $s=8$. The values represent the KL divergence (averaged over 10 independent trails) between the empirical distribution of the attention output ($y_{1}^{1}$) and our theoretical limit ($Z^{y^{1}}$), corresponding to the analysis in Figure 1(b).
>
> | $n$ | 16 | 64 | 256 | 1024 |
> |---|---|---|---|---|
> | log-mean | -2.78 | -5.02 | -6.68 | -7.31 |
> | log-mean - 1SD | -2.93 | -5.22 | -6.89 | -7.58 |
> | log-mean + 1SD | -2.63 | -4.82 | -6.48 | -7.03 |
>
> As the table demonstrates, the predictions of our theory remain highly accurate even when the number of tokens is changed. We plan to conduct further experiments with different activation functions in the future. These new results will be incorporated into the appendix of our revised manuscript.
>
> **Question: Is there a way you can make the notation more readable?**
>
> **Reply:** We agree that the notation, while consistent with that used in Yang (2020), may pose a burden for readers. To improve clarity, we will include a notation summary table in the appendix and clarify key notational conventions when first introduced in the main text. We hope this will make the paper more accessible and easier to follow, especially for readers new to this line of work.
>
> Furthermore, to improve clarity and make the notation more accessible, we will add a pseudocode example for the multi-head attention layer in the style of the NETSOR programs from Yang (2019) in the appendix of our revised paper.
>
> **Reference**
>
> [1] Yang (2019). Tensor programs i: Wide feedforward or recurrent neural networks of any architecture are Gaussian processes.
>
> [2] Yang (2020). Tensor programs iii: Neural matrix laws.
>
> [3] Yang and Hu (2021). Tensor programs iv: Feature learning in infinite-width neural networks.

---

### Official Review · Reviewer_pKBC · 2025-07-03

**Clarity:** 2
**Significance:** 2
**Originality:** 3
**Rating:** 4
**Confidence:** 2

**Summary:**

The paper studies the infinite-width limit of a single multi-head attention layer under a finite number of heads and the standard $1/\sqrt{n}$ scaling for attention. The authors rigorously prove that under these conditions, the limiting distribution of the layer's output is non-Gaussian. They characterize this distribution as a hierarchical Gaussian, where the output is Gaussian conditional on the random attention scores, which themselves converge to a Gaussian distribution.

**Questions:**

- The core result is that the limiting distribution is non-Gaussian, which breaks the standard analysis for deep models. Do the authors have any insights, even speculative, on how this challenge might be overcome? For example, are there frameworks or modifications to tensor programs that could handle such hierarchical Gaussian distributions as inputs for subsequent layers?
- What are the main differences between the hierarchical Gaussian distribution and the standard Gaussian? How might this inherent non-Gaussianity affect signal propagation (e.g. rank collapse), feature learning, or the optimization landscape in self-attention models?
- To improve readability, the authors may consider adding a pseudocode example for a single attention layer, in the style of the Netsor programs from [Yan19b]. This would be helpful for readers.

**Ethical Concerns:**

["NO or VERY MINOR ethics concerns only"]

**Final Justification:**

The paper provides a rigorous analysis of the infinite-width limit of a realistic attention setting (finite heads, standard scaling), addressing a gap in prior work. While the breakdown of Gaussianity is somewhat expected, the precise characterization of the hierarchical Gaussian distribution is a meaningful contribution. The main limitation is that the implications, both theoretical and practical, are discussed only speculatively, without concrete analysis or validation. Nevertheless, the authors acknowledge this and propose future directions. Overall, I think the paper offers valuable theoretical insights that warrant a borderline accept.

**Limitations:**

Yes

**Quality:**

3

**Strengths And Weaknesses:**

**Strengths:**
- The paper's main strength is its focus on a realistic setting for attention ($1/\sqrt{n}$ scaling, finite heads), which has been a gap in prior infinite-width analyses. This work moves beyond the simplifying assumptions of infinite heads or $1/n$ scaling to provide a more faithful theoretical model.
- The mathematical derivation, leveraging tensor programs, is rigorous and technically sound, with numerical experiments supporting the theoretical results.

**Weaknesses:**
-  The paper proves the non-Gaussianity but does not discuss what this means for the behavior of attention layers or deep Transformers. A deeper discussion of these implications is missing, which limits the paper's overall contribution to our understanding of these architectures (see question 2).
- The paper demonstrates that the Gaussianity breaks down at the very first attention layer but offers no discussion or potential solution for how to proceed with the analysis of deeper models. While rigorously proven, this result is somewhat expected by the community ([HBSDN20]).
- While the paper is generally well-organized, the mathematical notation is dense and can be difficult to follow.

---

> ### Author Rebuttal · Authors · 2025-07-31
>
> Thank you for the constructive feedback and for highlighting our contribution.
>
> **Weakness 2-2: The non-Gaussianity is somewhat expected.**
>
> **Reply:** We agree that the breakdown of Gaussianity in attention mechanisms with finite heads has been heuristically observed in earlier studies, particularly in Hron et al. (2020), which provides compelling empirical evidence and predictions. Our work builds on this intuition by rigorously establishing the exact form of the limiting distribution for a single attention layer under realistic settings (finite heads and standard $1/\sqrt{n}$-scaling). In this sense, while the emergence of non-Gaussianity may be expected qualitatively, we believe that the precise derivation of the hierarchical Gaussian structure and its formal validation is a meaningful contribution to the theory of attention.
>
> **Q1: Analysis of deeper models and overcoming the non-Gaussian challenge.**
>
> **Reply:**  This is an excellent and challenging question. We predict that the behavior of the attention outputs will likely deviate even further from Gaussianity as layers are stacked. In fact, our analysis indicates that even at the first layer, the output already exhibits heavier tails than a Gaussian. This suggests that as layers accumulate, the output distribution may increasingly diverge from any hierarchical Gaussian form, possibly yielding even heavier-tailed or more structured non-Gaussian behaviors.
>
> While a complete analysis of deep attention networks remains an open and challenging direction, we believe our work offers a necessary first step toward that goal. To our knowledge, a theoretical framework for handling such evolving non-Gaussian distributions does not yet exist. We are currently exploring how the hierarchical structure evolves under layer composition, and we view this as an exciting avenue for future research.
>
> We thank the reviewer again for highlighting this important point. We will incorporate a discussion of this important challenge into the revised version of our paper.
>
> **Q2: Implications of non-Gaussianity and differences from a standard Gaussian.**
>
> **Reply:** We agree that a deeper discussion on the implications of our findings is crucial.
>
> The hierarchical Gaussian distribution derived in our work differs from the standard Gaussian in a fundamental way: it introduces randomness in the variance itself. That is, the output distribution is conditionally Gaussian given a set of random similarity scores, but these scores themselves follow a Gaussian distribution, resulting in a scale mixture of Gaussians. It is well-known in probability theory that such hierarchical structures produce heavier tails than standard Gaussians. For instance, when the variance follows a uniform distribution, the resulting distribution is a normal scale mixture; when it follows the square of a Gaussian, the resulting distribution becomes a variance-gamma distribution, both of which are strictly heavier-tailed than the Gaussian.
>
> This inherent non-Gaussianity may significantly affect several aspects of learning in self-attention models. For example:
> 1. In feature learning, the alignment of activations with specific vector directions is often critical. Heavier-tailed distributions may make such alignment easier by amplifying high-magnitude components, thereby facilitating the emergence of dominant features.
> 2. In terms of optimization landscapes, the anisotropy induced by non-Gaussianity could lead to more irregular curvature, possibly affecting convergence properties of training dynamics (e.g., through sharper gradients or more prominent saddle regions).
> 3. Additionally, signal propagation may be influenced by the presence of higher-order moments. For instance, heavier tails might reduce the risk of rank collapse by preserving variability in feature representations across layers.
>
> At the same time, the non-Gaussian nature of attention outputs poses significant challenges for theoretical analysis. Many existing frameworks, such as Tensor Programs, rely heavily on Gaussian assumptions for tractability. Extending these tools (or developing new ones) to rigorously capture non-Gaussian behaviors is an intriguing and important open question, and we believe our work provides a concrete starting point for such future developments.
>
> We appreciate the reviewer’s insight in drawing attention to these broader implications, and we hope our response clarifies the potential impact and theoretical interest of the non-Gaussian structure revealed in our study.
>
> **Q3: Readability and pseudocode.**
>
> **Reply:** We appreciate this suggestion. To improve clarity and make the notation more accessible, we will add a pseudocode example for the multi-head attention layer in the style of the NETSOR programs from Yang (2019) in the appendix of our revised paper.
>
> **Reference**
>
> [1] Hron et al. (2020). Infinite attention: NNGP and NTK for deep attention networks.
>
> [2] Yang (2019). Tensor programs i: Wide feedforward or recurrent neural networks of any architecture are Gaussian processes.

---

> > ### Comment · Reviewer_pKBC · 2025-08-06
> >
> > Thank you for the detailed clarifications. I appreciate the authors’ responses and their transparency regarding current limitations. I will keep my score, as the paper remains technically solid.

---

### Note · Authors · 2025-08-14

We thank the reviewers and AC for their careful evaluations and constructive dialogue. This note provides closure by clarifying our contributions.

**Setup and Contributions**. Our paper rigorously characterizes the infinite-width limit of a single attention layer under the realistic $1/\sqrt{n}$-scaling with a finite number of heads, showing that the output follows a hierarchical Gaussian law, with experiments validating the theory at finite width. This fills the gap between attention layers in practice and the prior results for attention layers with $1/n$–scaling and the infinite-heads based on Gaussian approximation.

Through productive reviews and discussions, the contribution's key strengths and its impact became clearer. Below we summarize the main points that, in our view, firmly establish its significance.
- **Novelty:** We provide, to our knowledge, the first rigorous treatment of a finite-head, $1/\sqrt{n}$-scaling attention layer at infinite width, clarifying why the limit is non-Gaussian and how its hierarchical structure arises. This result fills the gap between a realistic design of attensions and the existing tensor-program results/infinite-head approximations.
- **Realistic setting:** Reviewers highlighted the focus on finite heads with $1/\sqrt{n}$-scaling, which matches the scaling used in practice.
- **Rigor and clarity:** Reviewers confirmed that the derivation via Tensor Programs is sound and self-contained. In response to clarity suggestions, we will add (i) concise pseudocode in the NETSOR style and (ii) a compact notation summary.
- **Empirical support** Finite-width simulations align closely with theory, including checks across token counts and practically relevant low-rank settings.
- **Broader implications:** The hierarchical, heavier-tailed structure suggests anisotropy and deviations from Gaussianity that may compound when stacking layers, pointing to concrete directions for multilayer and training-time analyses.
- **Related to other regimes:** Following the suggestion to include the $1/\sqrt{H}$ factor, our updated comparisons show our finite-head non-Gaussian law approaches the Gaussian infinite-head limit. We also clarify relations to $\mu P$ scalings and kernel/DMFT perspectives.

We appreciate the constructive tone of the discussion. We hope these clarifications and concrete revisions help the AC in the decision process.

---

### Decision · Program_Chairs · 2025-09-17

**Decision:**

Accept (poster)

**Comment:**

This paper gives a rigorous characterization of the infinite-width limit of a single attention layer under the standard scaling with a finite number of heads. Using Tensor Programs, the authors prove that the layer’s outputs converge to a hierarchical Gaussian law. Finite-width simulations (including variations in token count and low-rank settings) closely match theory. Together with comparisons to infinite-head regimes, the result cleanly fills a gap between prior analyses relying on infinite heads or nonstandard scalings and the setting used in real Transformers.

Reviewers broadly agreed the problem and setting are important and the mathematics sound. Strengths highlighted include: (i) rigorous limit law; (ii) technically careful derivations within Tensor Programs; and (iii) empirical checks that substantiate the limit in realistic ranges. Main weaknesses are scope and presentation: the result is for one layer and the forward pass at initialization only; implications for deep models and training dynamics remain largely speculative; notation is heavy; prior relevant work is not thoroughly discussed; and experiments are modest. During rebuttal the authors (a) clarified relations to infinite-head limits and adopted an explicit 1/√h factor to make head-count scaling comparisons fair, showing convergence toward the Gaussian infinite-head law; (b) added discussion of consequences of non-Gaussianity (heavier tails/anisotropy) and how this might propagate with depth; (c) provided additional experiments varying the number of tokens; and (d) committed to clearer presentation and to broadening related-work coverage. All reviewers voted for acceptance. I recommend acceptance as well.